
# Limitations of WRF land surface models for simulating land use and land cover change in Sub-Saharan Africa and development of an improved model (CLM-AF v. 1.0)

Timothy Glotfelty[1], Diana Ramírez-Mejía[2], Jared Bowden[3], Adrian Ghilardi[2], and J. Jason West[1]

[1]Department of Environmental Sciences and Engineering, University of North Carolina at Chapel Hill, Chapel Hill, NC 27599, USA

[2]Centre for Research in Environmental Geography, Universidad Nacional Autónoma de México, Morelia, 58190, Mexico

[3]Department of Applied Ecology, North Carolina State University, Raleigh, NC 27695, USA

*Correspondence to*: Timothy Glotfelty (twglotfe@email.unc.edu)

**Abstract.** Land use and land cover change (LULCC) impacts local and regional climates through various biogeophysical processes. Accurate representation of land surface parameters in land surface models (LSMs) is essential to accurately predict these LULCC-induced climate signals. In this work, we test the applicability of the default Noah, Noah-MP, and CLM LSMs in the Weather Research and Forecasting Model (WRF) over Sub-Saharan Africa. We find that the default WRF LSMs do not accurately represent surface albedo, leaf area index, and surface roughness in this region due to various flawed assumptions, including the treatment of the MODIS woody savanna LULC category as closed shrubland. Consequently, we developed a WRF CLM version with more accurate African land surface parameters (CLM-AF), designed such that it can be used to evaluate the influence of LULCC. We evaluate meteorological performance for the default LSMs and CLM-AF against observational datasets, gridded products, and satellite estimates. Further, we conduct LULCC experiments with each LSM to determine if differences in land surface parameters impact the LULCC-induced climate signals. Despite clear deficiencies in surface parameters, all LSMs reasonably capture the spatial pattern and magnitude of near surface temperature and precipitation. However in the LULCC experiments, inaccuracies in the default LSMs result in illogical localized temperature and precipitation climate signals. Differences in thermal climate signals between Noah-MP and CLM-AF indicate that the temperature impacts from LULCC are dependent on the sensitivity of evapotranspiration to LULCC in Sub-Saharan Africa. Errors in land surface parameters indicate that the default WRF LSMs considered are not suitable for LULCC experiments in tropical or Southern Hemisphere regions, and that proficient meteorological model performance can mask these issues. We find CLM-AF to be suitable for use in Sub-Saharan Africa LULCC studies, but more work is needed by the WRF community to improve its applicability to other tropical and Southern Hemisphere climates.

## 1 Introduction

Land use and land cover change (LULCC) has various biogeophysical impacts on climate by altering land surface albedo, evapotranspiration, and surface roughness that in turn alter atmospheric circulations, energy budgets, and hydrologic cycles



(Pielke et al., 2011; Mahmood et al., 2014; Bright 2015; Smith et al., 2016; Quesada et al., 2017). Results from global modelling studies indicate a global reduction in surface temperatures due to deforestation, but the impacts of LULCC vary by region and season (e.g., Zhao and Pitman 2002; Lamptey et al., 2005; Lejune et al., 2017). Such studies have shown a latitudinal difference in the temperature response to deforestation, where higher latitudes experience cooling in winter as less tree cover

brightens the surface when snow is present, and lower latitude tropical regions experience warming in response to a reduction in evaporation (e.g., Longobardi et al., 2016; Quesada et al., 2017). This LULCC latitudinal dependence has been shown to occur in observations as well (Zhang et al., 2014).

Impacts of LULCC are simulated in climate and numerical weather prediction models through the land surface model (LSM). Differences in LSM parameterizations can lead to significantly different climate responses to LULCC in both magnitude and

sign (e.g., Olsen et al., 2004; Boisier et al., 2012; Burakowski et al., 2016), even when little difference exists in the mean simulated climate (Crossly et al., 2000). Errors and uncertainties in LSMs occur in response to errors in LULC classification maps and the way land use properties, such as vegetation distributions and surface albedo, are prescribed (e.g., Lu and Shuttleworth, 2002; Olsen et al., 2004; Ge et al., 2007; Boisier et al., 2012; Boisier et al., 2013; Boysen et al., 2014; Meng et al., 2014; Hartley et al., 2017; Bright et al., 2018). As a result, improving LULC maps and LSM parameters has been shown

to significantly reduce biases and errors within global and regional climate models (RCMs) (e.g., Tian et al., 2004b; Kang et al., 2007; Lawrence and Chase, 2007; Lawrence and Chase, 2009; Moore et al., 2010; Karri et al., 2016; Thackeray et al., 2019). Having accurate representations of these parameters is especially important in regions with large surface heterogeneity, such as East Africa (Ge et al., 2008).

Sub-Saharan Africa is a region of particular interest for simulating LULCC because it has already experienced dramatic

LULCC (e.g., Collier et al., 2008), which has been shown to alter the West African monsoon system (e.g., Charney, 1975; Xue and Shakula, 1993; Abiodun et al., 2008; Wang et al., 2017). Various ensembles of RCMs have been applied to study the climate of Africa as part of both the COordinated Regional climate Downscaling Experiment (CORDEX) (e.g., Nikulin et al., 2012; Gbobaniyi et al., 2014; Kim et al., 2014; Mounkaila et al, 2015; Endris et al., 2016; Diasso and Abiodun, 2017; Adeniyi and Dialu, 2018; Odoulami et al., 2019) and the West African Monsoon Modeling and Evaluation Project Experiments

(WAMME) (e.g., Wang et al., 2016; Xue et al., 2016). Included as part of these ensemble modelling projects is the Weather Research and Forecasting (WRF) Model (e.g., Xue et al., 2016; Fita et al., 2019).

The WRF model is a state-of-the-art numerical weather prediction model designed to be applicable in multiple world regions, across multiple spatial scales, and for short-term forecasting to longer term regional climate simulations (Skamarock and Klemp, 2008). Multiple studies have tested the sensitivity of the African climate to different ensembles of WRF physics

parameterizations, including LSMs (e.g., Pohl et al., 2011; Hagos et al., 2014; Noble et al., 2014; Alaka and Maloney, 2017; Noble et al., 2017; Igri et al., 2018). Results from these WRF simulations are somewhat contradictory as some studies found the National Centers for Environmental Prediction, Oregon State University, Air Force, Hydrology Lab (Noah) LSM (Chen and Dudhia, 2001; Ek et al., 2003) to have superior performance compared to observations and reanalysis (Pohl et al., 2011; Igri et al., 2018), while others found no unambiguous difference in model performance between different LSMs (Noble et al.,





2014; 2017). In terms of LULCC applications, Hagos et al. (2014) found that WRF model configurations that simulate a climate which is too wet or too dry compared to observations and reanalysis do not produce a strong climate signal from LULCC over Africa. This weak signal is a result of the model falling into erroneous moisture or energy limited regimes. Despite these uncertainties, the Noah LSM is by far the most common LSM configuration applied in WRF studies over Africa (e.g., Vigaud et al., 2011; Cretat et al., 2012; Boulard et al., 2013; Ratna et al., 2014; Schepanski et al., 2014; Argent et al.,

2015; Diaz et al., 2015; Klein et al., 2015; Zheng et al., 2015; Arnault et al., 2016; Kerandi et al., 2017; Klein et al., 2017). In this work, we expand upon the current literature by testing five different LSM configurations within the WRF model for the purpose of evaluating the effects of LULCC over time on regional climate in Sub-Saharan Africa. First, we review four commonly used LSMs to determine if the LSM configurations reasonably represent land surface parameters such as albedo and leaf area index (LAI). As shown below, we find that these four LSMs have significant deficiencies which limit their

capabilities in applications to LULCC in this region. Consequently, we then detail how we modify one LSM for use in this study. We then evaluate the five WRF LSM configurations against available meteorological observations, reanalysis, and satellite estimates to determine how well they simulate the current climate of Sub-Saharan Africa. Finally, we simulate the effects of LULCC over time on the simulated regional climate, and how these climate responses differ when using different LSMs. Understanding the deficiencies in how LSMs represent LULCC is key to accurately representing regional climate

signals that impact not only climate change investigations, but also coupled natural and human system research regarding human decision making, air quality, and human/ecosystem health interactions.

**2 WRF description and configurations**

This study uses the WRF model version 3.9.1.1 (WRFv3.9.1.1), configured as shown in Table 1, to simulate the regional meteorology and climate within Sub-Saharan Africa. We define a Sub-Saharan Africa domain that ranges from ~19º N - 35º

S latitude and ~19º W - 64º E longitude (Fig. 1) with a horizontal grid spacing of 36 km and 30 vertical layers from the surface to 50 hPa. Physics parameterizations common to all simulations include: the New Tiedtke cumulus parametrization scheme (Zhang et al., 2011), the aerosol-aware Thompson microphysics scheme (Thompson and Eidhammer, 2014), the RRTMG long and shortwave radiation schemes (Clough et al., 2005; Iacono et al., 2008), and the MYNN surface/ planetary boundary layer physics (Nakanishi and Niino, 2004; 2006). These physics combinations were selected because they represent some of the

most advanced science within the WRF model, and these physics options performed the best when validated against observations/satellite estimates relative to other physics options tested (not shown). All simulations also take advantage of the CLM4.5 lake model, which is calibrated to prognostically simulate lake conditions for the African Great Lakes by adjusting the lower bound lake temperature from 4°C to 24°C consistent with Lake Victoria temperature profiles (Nyamweya et al., 2016). Meteorological initial and boundary conditions for the simulations are obtained from the European Centre for Medium-

Range Weather Forecasts Interim reanalysis (ERA-Interim) (Dee et al., 2011). Because LULC inputs change each year, each





model year is modelled individually, preceded by a three-month spin-up period that is discarded to allow the model to reach equilibrium and minimize the impact of initial conditions on the simulations.

## 2.1 WRF land surface model descriptions

Here we briefly describe four commonly-used WRF LSM configurations used in this study and differences between them: the
Noah LSM; the Noah LSM using satellite derived albedo and LAI (Noah-Sat); the Noah Multi-Parameterization LSM (Noah-MP) (Niu et al., 2011); and the default Community Land Surface model (CLM-D) (Subin et al., 2011; Jin and Wen, 2012; Lu and Kueppers, 2012). We focus on the different ways in which the LSMs prescribe and treat surface parameters such as LAI, albedo, and surface roughness length based on the Moderate Resolution Imaging Spectroradiometer (MODIS) 21 land category data. In addition to the LSMs used in this work, four other LSMs exist in WRF including: the five layer thermal diffusion
scheme (Skamarock et al., 2008), the Rapid Update Cycle (RUC) LSM (Smirnova et al., 2016), the Pleim-Xiu (PX) LSM (Pleim and Xiu, 2003; Gilliam et al., 2007), and the Simplified Single Biosphere Model (SSiB) (Xue et al. 1991; Sun and Xue, 2001). The five layer thermal diffusion scheme is omitted from these experiments because it is overly simplistic and not appropriate for climate scale studies. The RUC and PX LSMs are primarily designed for weather forecasting and for retrospective meteorological simulations commonly used as input for downstream air quality simulations, respectively.
Although RUC and PX can be used for other applications, they require extensive detailed input data or data assimilation for peak performance (http://www2.mmm.ucar.edu/wrf/users/docs/user_guide_V3.9/users_guide_chap5.htm#Phys). Since this detailed level of observational data is not available in Sub-Saharan Africa, both the RUC and PX scheme were excluded. The SSiB LSM is designed for climate applications, however it is also excluded both because its best performance occurs using its own LULC dataset and because it is not currently compatible with the MYNN surface/boundary layer parameterizations.

### 2.1.1 Noah LSM and Noah-Sat

In Noah and Noah-Sat are the same LSM with different configurations for how surface albedo and LAI are prescribed. Within the Noah LSM, surface parameters including surface albedo, roughness length, and LAI are prescribed based on the dominant MODIS LULC category in each grid cell with temporal interpolation between maximum and minimum values depending on the time of year. The Noah-Sat configuration uses a monthly average satellite derived climatology of surface albedo and LAI
supplied from the WRF preprocessing system (WPS), as a more detailed replacement of the LULC based prescribed values. Noah and Noah-Sat have no explicit canopy layer, and instead simulate evapotranspiration using a satellite derived green fraction variable from WPS to weight the contribution of direct soil evaporation and evapotranspiration from vegetation in each grid cell. The land surface and underlying soil is simulated using 4 soil layers 0.1, 0.3, 0.6, and 1.0 meters thick centered 0.05, 0.25, 0.7, and 1.5 meters below the ground surface, respectively.
Noah-Sat is limited in its ability to simulate LULCC because LAI and surface albedo are decoupled from changes in the LULC categories, and temporally varying satellite LAI and albedo products are influenced by other climate variations or changes apart from effects of LULCC. Noah is also preferable to Noah-Sat for future climate simulations because the albedo and LAI





products Noah-Sat requires would have to be generated as separate independently varying fields from the future LULC projection.

Additionally, the Noah LSM can be configured using a mosaic tile approach to represent the influence of sub-grid scale variations in LULC. The representation of sub-grid LULC variability can significantly alter the responses of climate models to LULCC (e.g., Boone et al., 2016), but this functionality is not considered in these experiments since any underlying errors in albedo, LAI, and surface roughness from Noah would be present in both the mosaic tile and dominant LULC configurations. Also, this approach has been shown to primarily impact urban regions (Mallard and Spero, 2019), which are not resolved well

at the grid spacing of this study.

### 2.1.2 Noah-MP

The Noah-MP model is an updated version of the Noah LSM with multiple-parameterization options utilizing the same soil level structure as the default Noah LSM. The major updates in Noah-MP include: the addition of an explicit one-layer vegetation canopy and three layer snowpack, a tiling scheme that separates vegetation and bare soil to better calculate the

surface energy balance, separating permeable and impermeable frozen soils, new runoff and groundwater schemes, and new dynamic vegetation model options (Niu et al., 2013; Xia et al., 2017 and references therein). In this study, Noah-MP is configured with the default settings, which are the most similar to the default Noah LSM. With these default settings, dynamic vegetation is disabled and LAI is prescribed based on the dominant MODIS LULC category in each grid cell using monthly profile values. Noah-MP simulates surface reflectance using a modified two stream radiation scheme that accounts for gaps

within the vegetation canopy and between canopy crowns (Yang and Friedl, 2003; Niu and Yang, 2004); however, in WRFv3.9.1.1 Noah-MP uses a simplification that assumes all bare soil albedos are comparable to loam soil. As a result, surface albedo within Noah-MP is solely a function of soil moisture and vegetation cover.

### 2.1.3 Default CLM (CLM-D)

The default configuration of CLM in WRF divides the land surface into five types: glacier, lake, wetland, urban, and vegetated.

Vegetated land is further split into up to four patches of 16 plant functional types (PFTs) with distinct physiological parameters. Calculations within each vegetated grid are done at the PFT level and then aggregated for atmosphere interactions. CLM contains a single-layer vegetation parametrization with a sunlit and shaded canopy and uses the two stream approximation (Sellers 1985) to calculate the energy balance within the canopy. Temperature and humidity varies between the ground surface, the canopy, and the leaf surface (Subin et al., 2001 and references therein). The land surface and soil properties in CLM are

simulated using 10 layers ~0.018, 0.028, 0.045, 0.075, 0.124, 0.204, 0.336, 0.554, 0.913, and 1.134 meters thick centered at ~0.007, 0.028, 0.062, 0.119, 0.212, 0.367, 0.620, 1.038, 1.728, and 2.86 meters below the ground surface.

In the version of CLM available in WRF, each dominant MODIS land use category is assigned a distribution of PFTs with distinct monthly profiles for LAI that do not vary geographically. A list of the CLM PFTs with the percentages for each



vegetated MODIS land use category is shown in Table S1 of the Supplementary Material. Bare soil albedos in CLM are not

constrained like within Noah-MP and therefore a broader range of surface soil albedos is considered.

Some simplifications in WRF-CLM lead to difficulties applying the default version for the Sub-Saharan Africa domain. For example, Table S2 of the Supplementary Material shows the monthly LAI profiles used for each PFT within the default CLM configuration. These profiles clearly show Northern Hemisphere growing cycles, which is problematic for Sub-Saharan Africa because it contains regions with bimodal tropical growing cycles and Southern Hemisphere growing cycles. Additionally, the

visible spectrum dry soil albedo for the sandiest soils in the default CLM treatment is 0.24, considerably less than the 0.25-0.45 albedo from MODIS satellite estimates over most the Sahara (Wang et al., 2004)..

## 3 Updated CLM for Sub-Saharan Africa (CLM-AF)

To address these limitations with CLM-D, and deficiencies of other LSMs described in the results section, the WRF-CLM LSM has been modified to include PFT distributions more representative of the Sub-Saharan Africa domain, regionally varying

monthly profiles for LAI and stem area index (SAI), minor improvements in vegetation optical properties (e.g., leaf reflectance), and scaled surface albedos for sandy soils to better match satellite estimates. Each of these modifications is described in detail below.

### 3.1 CLM-AF plant functional type distributions

Updated PFT distributions are derived from a global three arc minute PFT dataset for the year 2001 generated by the National

Center for Atmospheric Research for the Model of Emission of Gases and Aerosols from Nature version 2.1 (https://bai.ess.uci.edu/megan/versions/megan21). To determine the percentages of each PFT representative of the various MODIS land use categories in Sub-Saharan Africa, the global PFT dataset is regridded to the 36 km WRF domain, and the average coverage of each PFT within each WRF-MODIS 2001 dominant land use category is calculated. Updated PFT distributions were generated for broad leaf evergreen/deciduous forests, mixed forests, closed and open shrublands, woody

savannas/savannas, grasslands, and cropland/mosaic croplands (i.e., MODIS categories 2,4,5, 6-10, 12, and 14). This limited subset of categories is used because the remaining MODIS categories did not cover a large enough area to be the dominant land use at 36 km resolution. Since CLM allows for up to four PFT patches, the top four most abundant PFTs within each MODIS land use category are scaled to represent one hundred percent of the land use category, with an exception for some inconsistencies that occurred between the PFT and the evergreen broad leaf forest, savanna, and mosaic cropland MODIS

categories (see Supplementary Material). The resulting updated PFT distribution for these CLM vegetated land use categories is shown in Table 2.

Most of the updated PFT distributions in Table 2 are consistent with the MODIS International Geosphere-Biosphere Programme (IGBP) category descriptions (Table S3, Supplementary Material). However, there are two minor inconsistencies with the closed shrubland and grassland categories. The closed shrubland category contains slightly less than 60% shrubs and





8% deciduous tropical trees, indicating there is some sub-grid scale overlap with nearby woody savannas or forests. The grasslands category contains 18% shrubs, which is higher than the 10% from the description in Table S3, indicating some overlap with sub-grid scale shrublands. Overall, compared to the CLM-D PFT distributions in Table S1, the updated values in CLM-AF have greater heterogeneity in plant types and contain more herbaceous cover. The largest deviations from the CLM-D distribution occur with shrublands and woody savanna. CLM-D prescribes all shrublands as broad leaf evergreen shrubs,

while the global PFT dataset indicates that shrublands in Sub-Saharan Africa contain broad leaf deciduous temperate shrubs. Additionally, the woody savanna category PFT distribution in CLM-D is identical to closed shrubland. This is potentially a large source of error as woody savanna should have forest cover between 30-60% (Table S3, Supplementary Material). This error is removed in the CLM-AF PFT distribution with the woody savanna category containing 38% tree cover.

### 3.2 CLM-AF LAI and SAI profiles

Since the Sub-Saharan Africa domain covers a wide range of tropical and sub-tropical latitudes, a single domain-wide LAI and SAI monthly profile for each PFT is not appropriate. Here, geographically varying monthly LAI profiles are generated by dividing the Sub-Saharan Africa domain into 17 distinct regions based on bioclimate characteristics used in LULCC modelling of Sub-Saharan Africa (Fig.1, Table 3).These bioclimate regions were generated because landscape dynamics are known to be different between broad climatic zones, needing separate modelling parametrizations (Soares-Filho et al 2006).  However, the

central wet (CW), central moist (CM), and northeast semi-dry (NESD) bioclimate regions span a large latitudinal range and are subdivided based on latitude to generate more meaningful LAI seasonal profiles (Supplementary Material).

The updated LAI profiles within each bioclimate region are derived from both the 36 km regridded global PFT dataset and the monthly LAI climatology data, provided by WPS, used in the Noah-Sat configuration. LAI profiles are calculated only from a subset of grid cells within the WPS Sub-Saharan Africa LAI climatology, where the 36 km regridded PFT data indicates that

a given PFT comprises eighty percent or more of the grid cell (PFT80). For the broad leaf evergreen tree PFTs, the median monthly LAI value of the PFT80 grid cells within each bioclimate or sub-bioclimate region is used as the monthly prescribed LAI value for that PFT. Median values are used in place of mean values for the broad leaf evergreen tree PFTs because several small LAI values near the edges of forested regions lead to unrealistically small LAI values for the Congo and other forests compared to the WPS LAI satellite derived climatology.  For the remaining PFTs, the mean monthly LAI value of the PFT80

grid cells within each bioclimate or sub-bioclimate region is used as the monthly prescribed LAI value for that PFT. The monthly LAI profiles for each PFT within each bioclimate and sub-bioclimate region are listed in Tables S4-S10 of the Supplementary Material. If no grid cells within a bioclimate or sub-bioclimate region meet the PFT80 criteria for a required PFT, then a reduced threshold of sixty percent of the grid cell is utilized to calculate the monthly LAI profile for that PFT. If no grid cells meet the sixty percent criteria, the LAI profile for that PFT within the bioclimate or sub-bioclimate region is

assumed to be the same as a nearby comparable bioclimate region. These comparable "alternative" bioclimate regions are listed in Table 4.  Some additional adjustments were also required for the broad leaf evergreen tree PFTs to make these areas more consistent with the satellite derived climatology (Supplementary Material).



SAI represents the area of stems and dead leaves. The values of SAI are poorly known, but SAI is generally parameterized to have a minimum in winter and maximum in autumn for each land cover type (Tian et al., 2004a). Since no readily available

data on SAI exists, SAI within CLM-AF is based on relating decreases in LAI (ΔLAI) from month to month to the SAI values in the CLM-D configuration. This is done by fitting a simple linear regression between ΔLAI and the SAI value in CLM-D. If the LAI is not decreasing from the previous month then the SAI value is assumed to be the minimum value from CLM-D. These assumptions are consistent with the definition of SAI as dead leaves/litter will only increase when LAI is decreasing. However, it was not possible to generate linear regressions for evergreen trees and corn from CLM-D because the evergreen

tree LAI profiles in CLM-D do not change from month to month and the corn SAI profile is equivalent to the corn LAI profile. These assumptions are not appropriate for Sub-Saharan Africa because of the longer tropical growing season and small seasonal fluctuations in evergreen tree LAI in the satellite climatology. Therefore, corn within CLM-AF is assumed to have the same SAI profile as C4 grass, and evergreen trees follow a similar equation as C4 grass with an intercept equivalent to the appropriate evergreen tree minimum SAI value of 0.5. A list of the SAI profile equations and minimum SAI values in the

CLM-AF configuration for the updated PFTs are listed in Table 5.

### 3.3 CLM-AF sandy soil albedo

CLM-D sandy soil albedos and updated values for CLM-AF are listed in Table 6. CLM simulates surface albedo using a look-up table for different soil color classes with two different radiation streams that differentiate between saturated and dry soils. Albedo values in the sandy soils of the Sahara range from 0.25-0.45 (Wang et al., 2004), which is larger than the 0.24 dry

sandy soil albedo in CLM-D. Accordingly, we increased the albedo values for sand and sand-loam combination soil types by 0.1 and 0.02, respectively. This puts the sandy soil albedos inside the range expected for the Sahara, while not leading to excessively large albedos in the deserts of southern and eastern Africa.

### 3.4 CLM-AF vegetation property adjustments

In order to bring the albedo of vegetated areas into better agreement with the satellite climatology from WPS, several

adjustments are made to leaf/vegetation optical properties in CLM-AF. In CLM-D, shrubs in Sub-Saharan Africa are erroneously classified as broad leaf evergreen shrubs rather than temperate deciduous shrubs. In order to maintain a lower albedo for these African shrubs, the leaf transmittance, leaf angle, and leaf reflectance properties of the deciduous temperate shrubs are adjusted to match those of broad leaf evergreen shrubs. Additionally, the near-infrared leaf reflectance of all broad leaf tree species is lowered from 0.45 to 0.35 in CLM-AF, which is in better agreement with near-infrared leaf reflectance measured by unmanned aerial vehicle mounted hyperspectral imaging instruments over African forest canopies (Thomson et

al., 2018).



## 4 Experimental design

This study consists of two experiments (Table 7). The first is a model validation experiment to compare differences between the WRF LSM configurations and assess their model performance. The second is a LULCC experiment to determine if the
errors and uncertainties of each LSM lead to differences in their climate responses to LULCC.

### 4.1 Model validation experiment

The model validation experiment consists of five simulations conducted for the year 2013, each using one of the five LSM configurations discussed above. The year 2013 is selected because it is a neutral year for the El Niño Southern Oscillation. While a single year comparison does not yield climate relevant statistics, it is sufficient to demonstrate differences in the
meteorology between the five LSM configurations and the mechanisms responsible for these differences. The model validation simulations are conducted with default greenhouse gas concentrations and MODIS 21 class land use data. These default settings are chosen to illustrate the performance that can be expected from the publicly available WRF model.

### 4.2 Land use and land cover change experiment

The LULCC experiment simulates the period 2010-2015 and represents recent climate responses from LULCC since the year
2001. To accomplish this goal, the LULCC experiment consists of two simulations per LSM configuration. The first simulation for each LSM uses static LULC from the year 2001 for each simulated year (i.e., 2010-2015), hereafter referred to as LU01. The second uses dynamic LULC each simulated year, hereafter referred to as LUD. Differences between the LU01 and LUD simulations delineate the climate response to LULCC. These two simulations are carried out using the Noah, Noah-MP, CLM-D, and CLM-AF LSMs. Noah-Sat is excluded because LAI and albedo parameters derived from satellite data could be
impacted by climatological variability other than LULCC. The LULCC simulations utilize MODIS 21 class land use data that was processed by the Dinamica EGO land use modeling framework (Soares-Filho et al., 2002 – described in more detail below) and global average greenhouse gas concentrations for each simulation year from the National Oceanic and Atmospheric Administration's (NOAA) Earth System Research Laboratory (ESRL) Global Monitoring Division. In the LULCC experiments, each year is a discreet simulation with a 3-month spin-up in which the model LULC is updated at the start of
each year. This is necessary because the WRF modelling framework treats LULC as a static field.

### 4.3 Land use / land cover data

The LULC dataset for the LULC experiment is extracted from simulated annual maps of land use and land cover spanning 2001 to 2050. These maps are created by means of prospective landscape modelling techniques and while simulations contain some level of model error, this approach is used to reduce the impact of potential LULC misclassification errors and
uncertainties in the MODIS product that could propagate into the WRF model leading to "noisy" and inconclusive climate impacts.





LULC is simulated using the Dinamica EGO environmental modelling platform (Soares-Filho et al., 2002). Dinamica EGO is a modelling tool designed to construct simple or complex spatiotemporal models involving multiple transitions and iterations, dynamic feedbacks, sub-region approaches, and several spatial algorithms for the analysis and simulation of a wide variety of

dynamic LULCC phenomena. Dinamica EGO has been used for many applications (Bowman et al., 2012; Cheng et al., 2020; De Almeida et al., 2005; Ghilardi et al., 2016; Merry et al., 2009; Nepstad et al., 2009; Oliveira et al., 2019; Silvestrini et al., 2011; Soares et al., 2002; Soares et al., 2006; Soares-Filho et al., 2010; Thapa and Murayama, 2011).

For input to our LULCC simulations, we use the MODIS Land Cover Type product (MCD12Q1) consisting of a suite of datasets that provides global land cover maps at 500 meter spatial resolution and annual temporal coverage from 2001 to 2013,

and includes six different land cover classification schemes (Friedl and Sulla-Menashe, 2015; Friedl et al., 2010). This product is generated using an ensemble of supervised classification algorithms that uses MODIS Nadir BRDF-Adjusted Reflectance data as input (Schaaf et al., 2002). Specifically, we use the IGBP classification legend since the land cover data ingested by the WRF model (Skamarock, 2008) is based on MODIS-IGBP classification scheme. The MODIS Land Cover product has a post-process overall accuracy of 75% (Friedl et al., 2010).

Using this approach, we detected spurious changes that toggle yearly between classes such as woody savannas, savannas, or grasslands. To reduce this temporal noise, we apply a cell-based temporal mode filter that replaces cell values with the most frequently occurring LULC class selected from a moving but non-overlapping 3-year window; or alternatively, a 6-year window when no mode is found, and assigning No Data to the entire 12-year time series if still inconclusive. This filter preserves long lasting changes while drastically reducing short term changes between LULC classes. There is no edge

preservation because windows do not overlap in time, i.e. LULC classes can change for 2001 or 2012. Consequently, the year of a "true" LULC change can be shifted forwards or backwards by one year.

Prospective landscape models covering very large areas need to be regionalized, meaning that during the calibration period, explanatory variables and their spatial relationships with observed changes can be tuned separately to capture the heterogeneity of landscape dynamics. Regions do not represent "hard borders" in modelling results, as the amount of projected change and

the probability of change occurrences are not boxed-in within regions, but the proximate causes of observed change can be analysed separately.

Africa is regionalized into 18 regions based on climatic zones, demographic factors, and anthropogenic activity (Fig. S1) consisting of three overlapping layers: 1) United Nations geographic regions for Africa: Northern, Eastern, Southern, Western and Central (UNSD, 1999); 2) a bioclimate layer from the modified version of the Global Environmental Stratification (GEnS)

dataset (Metzger et al. 2013); and 3) residential sector emissions hotspots using DICE-Africa emissions (Marais and Wiedinmyer, 2016). The resulting 67 categories are generalized into the final 18 based on neighborhood. The process of generalization is done by comparing major change trends among regions, trying to avoid as far as possible the presence of separate regions with similar LULC dynamics. Of these 18 regions, 17 are used in the WRF modelling because the North Semi-dry region is outside the Sub-Saharan Africa domain.





LULC change rates by region are analysed by means of transition probability matrices, in order to quantify the amount of change in km$^2$ for each LULC change transition during the calibration period (2001 – 2007). Matrices are annualized and used to simulate expected annual LULC changes up to 2013 for validation purposes, and to 2050 for simulation purposes.

While transition matrices project the expected amount of LULC change into the future, they say nothing about where this change is likely to occur. For each meaningful transition, a map depicting the probability of that transition happening in the

future is built by means of analysing the spatial relationship between observed changes and a set of explanatory variables (Table S11, Fig. S2). Static and dynamic explanatory variables were related by means of conditional probabilities with the spatial occurrence of observed LULC changes for a subset of meaningful transitions during the calibration period. All maps were resampled to 1 km$^2$ resolution and projected to the Africa Albers Equal Area Conic coordinate system. Annual transition matrices (how much change is expected at each year) are integrated with annual probability maps (also generated for each

year) to produce simulated land cover maps.

To evaluate the accuracy of simulations within the present time, simulated annual maps outside the calibration period (2008-2013) are compared with the MODIS product for the same year, using a fuzzy-logic method (Hagen 2003) (Fig S3). This approach incorporates a moving window neighborhood context, since predicting the location of LULC transition at a pixel level is virtually impossible. The comparison is done between simulated and observed cells undergoing a certain LULC change

within the windows. To measure the spatial agreement between maps we used window sizes ranging from 1 to 9 cells (corresponding to spatial resolutions between 500 x 500 m and 4,500 x 4,500 m). For most transitions and regions, simulations correctly predict change within 4,500 x 4,500 m windows 50% and 75% of cases, which is among the range of reported results in other prospective modelling studies (e.g. Carlson et al., 2012; Soares et al., 2012; Soares et al., 2006; Thapa and Murayama, 2011; Yi et al., 2012).

**4.4 Model evaluation datasets and protocol**

A list of data used to evaluate WRF's meteorological performance are shown in Table 8, and the WRF model variables evaluated against these datasets are listed in Table 9. Surface meteorological and climate quantities are validated against both hourly surface observations from the National Climate Data Center's Integrated Surface Dataset (NDCD-ISD) and monthly average gridded estimates from the University of East Anglia's Climate Research Unit version 4.02 (CRU TS4.02) dataset

(Harris et al., 2014). Precipitation (PRE) is evaluated against CRU TS4.02, monthly average Global Precipitation Climatology Project (GPCP) estimates, and three-hour average Tropical Rainfall Measurement Mission (TRMM) estimates. Cloud fraction (CF) and precipitable water vapor (PWV) are compared against estimates from the MODIS Terra Aerosol Cloud Water Vapor Ozone level three product (MOD08_M3). Additionally, radiation balance variables are compared against satellite estimates from the Clouds and Earth's Radiant Energy System – Energy Balanced and Filled (CERES-EBAF) dataset.

To compare WRF to the gridded datasets, the WRF output is averaged to the appropriate temporal resolution and regridded to the native horizontal resolution of the gridded products to calculate performance statistics. Comparisons against NCDC-ISD are made for each hour by pairing the monitoring station values to the value of the WRF 36 km grid cell containing the





monitoring station. All comparisons with MOD08_M3 are done by averaging the WRF data during the daytime MODIS Terra

overpass times of Sub-Saharan Africa (i.e., 600 - 1200 UTC). Model performance is determined by calculating the mean bias

(MB) and normalized mean bias (NMB). Statistical quantities are calculated for the domain as a whole and each bioclimate

region to show regional variability in model performance. Additionally, spatial patterns for the simulations and observations

are shown.

## 5 Results - surface parameters

It is important to illustrate the different surface properties from each LSM, within WRF, using the same LULC because these

surface properties differences will drive simulated changes in the meteorology. Figs. 2-4 depict the surface properties for each

of the five LSM configurations from the 2013 evaluation simulations. Since the albedo and LAI of Noah-Sat are generated

from satellite estimates, these values can be considered similar to observations.

In general, all of the LSM configurations that prescribe albedo overpredict surface albedo in vegetated areas. However, the

Noah LSM severely overpredicts surface albedo throughout the entirety of the domain with albedo values ranging from ~20-

28% for regions containing woody savanna, savanna, and shrubland (Fig. 2b), compared to 10-20% over the same areas in

Noah-Sat (Fig. 2a). This is because Noah's prescribed albedo values for many of the MODIS-IGBP categories are significantly

larger than those derived from satellites (Table S12, Supplementary Material). The annual average surface albedos prescribed

by Noah-MP (Fig. 2c) and CLM-D (Fig. 2d) are similar in magnitude and spatial pattern, with overpredicted albedo in

vegetated areas and underpredicted albedo in arid regions. However, Noah-MP underpredicts surface albedo in the Sahara to

a greater extent due to the loam soil simplification (Sect. 2.2.2). The Noah, Noah-MP, and CLM-D LSM configurations all

contain errors where woody savanna and closed shrubland are treated as either identical or similar. In the Noah LSM, this

leads erroneously to the woody savanna regions having greater surface albedos than nearby savanna regions. In Noah-MP and

CLM-D, this leads to woody savannas erroneously having lower albedos than nearby broad leaf evergreen forests, because

shrubs are assumed to have a lower leaf reflectance than broad leaf trees. In general, CLM-AF is the closest match to the

satellite spatial pattern, despite differences in magnitude (Fig. 2e). The prescribed albedo values in CLM-AF improve the

representation of surface albedo in the arid regions of northern and eastern Africa, but the scaled values lead to overpredictions

in southern Africa. Vegetated regions also contain higher albedo values than the satellite estimates. These errors suggest that

better representations of soil color and leaf reflectance are needed in WRF-CLM.

In general, all of the LSM configurations that prescribe LAI overpredict the LAI of arid regions compared to satellite estimates

(Fig. 3). Due to the lack of geographically varying LAI in CLM-D, the seasonality of LAI in Sub-Saharan Africa in this

configuration is incorrect with elevated LAI values throughout the entire domain for June, July, and August (JJA) and

minimum LAI values throughout the domain in January, February, and December (JFD). Additionally, woody savanna (see

Fig. 10) LAI in CLM-D is significantly underpredicted because it has the same LAI profile as closed shrubland.





Unlike CLM-D, the Noah and Noah-MP configurations account for differences in seasonality in the northern and southern

hemispheres by shifting the northern hemisphere LAI profiles by six months for the southern hemisphere. This approach leads

to differentiated northern and southern hemisphere LAI values in Noah and Noah-MP (Fig. 3); however, distinct discontinuities

occur in LAI at the equator. In Noah-MP this LAI discontinuity only impacts East Africa due to the presence of broad leaf

evergreen forest with a time invariant LAI profile (category 2 in Table S13) in Central Africa. This issue is more apparent in

the Noah LSM as the LAI discontinuity occurs in both eastern and central Africa, since all the LULC categories in this region

have time variable maximum and minimum LAI values (Table S14 of Supplementary Material). Additionally, the LAI profiles

in Noah-MP (Table S13) have a stronger seasonality than the Noah values due to many LULC categories having much lower

minimum values of ~0.0-0.5 during the winter months. This leads an overall underpredicted LAI during the winter periods in

both hemispheres and overpredicted LAI during the summer periods. The net effect of this error is an overall underprediction

in the annual average LAI of tropical heavily vegetated regions and slightly overpredicted annual LAI in sub-tropical arid

regions. The generally higher minimum and maximum LAI values in the Noah LSM lead to generally accurate annual average

LAI values in tropical regions, but significantly overpredicted annual LAI values in sub-tropical arid regions. The errors in the

LAI profiles of Noah and Noah-MP likely occur because they have been developed mainly for application in the Northern

Hemisphere Mid-Latitudes.

CLM-AF, generates annual and seasonal average LAI spatial patterns that largely mimic the satellite estimates (Noah-Sat).

The use of LAI profiles prescribed in smaller regions has eliminated any large and obvious discontinuities and better represents

the latitudinal variability and seasonality in LAI compared to the other LSM configurations. CLM-AF slightly underpredicts

LAI values in the south-eastern portion of the domain and slightly overpredicts LAI near the Sahara. These errors likely result

from the lack of spatial heterogeneity that can be expected from a look-up table methodology.

An observational RL dataset is not available for comparison with model estimated RL. However, a comparison of the modelled

RL (Fig. 4) reveals several critical issues with the default representations. Despite having accurate LAI and surface albedo

from satellite estimates, the Noah-Sat configuration uses the same methodology as Noah to prescribe RL and therefore both

LSMs possess the same limitations. The values of RL in Noah and Noah-Sat are very low in comparison to other LSMs, with

a maximum value over forested regions of 0.5 m. This is inconsistent with the MODIS-IGBP evergreen broad-leaf forest

definition of canopies larger than 2 m (Table S3), indicating that both of these configurations likely underestimate RL.

Additionally, the spatial patterns in Noah, Noah-Sat, and CLM-D are all incorrect due to prescribing woody savanna regions

as having shrubland RL values. The Noah-MP and CLM-AF LSMs have the most realistic spatial patterns and magnitudes of

RL. The key differences are higher RL values for herbaceous land cover types in Noah-MP and larger maximum RL values

over forested regions in CLM-AF.



## 6 Results - 2013 meteorological evaluation

The primary meteorological variable impacted by surface albedo is the upwelling surface shortwave radiation flux at the Earth's surface (USRS), shown for comparison with CERES-EBAF estimates (Fig. 5). The annual average surface plots illustrate that the Noah-Sat configuration, with satellite albedo estimates, has the best agreement between simulated USRS and CERES-EBAF. The performance of the remaining LSMs follows their agreement with the satellite albedo climatology (Fig. 2), where CLM-AF has the best performance and Noah the worst. Model performance is further quantified using soccer plots

(Fig. 6) of domain-wide and African bioclimate region NMB and NME statistics for simulated USRS and T2 compared to CERES-EBAF and CRU/NCDC-ISD observations. These statistics confirm that Noah-Sat has the best overall USRS performance and that Noah significantly overpredicts USRS in nearly all regions with overpredicted surface albedo. The statistical performance of CLM-D, Noah-MP, and CLM-AF are similar in many African bio-climate regions, with CLM-AF generally having the best overall agreement. In particular, CLM-AF simulates USRS more accurately in the arid ND and ED

regions than both Noah-MP and CLM-D, which indicates that the increased sandy soil albedos in CLM-AF improve model performance. Additional radiative budget variables are evaluated against CERES-EBAF estimates in the supplementary material (Figs. S4-S6). We find that most other radiative parameters have minimal differences between LSMs, with most errors resulting from underestimated cloud radiative forcing consistent with other WRF experiments in Africa (e.g., Diaz et al., 2015). Interestingly, despite clear deficiencies in surface parameters and USRS in many of the LSMs, all LSMs reasonably capture

the spatial distribution and magnitude of annual T2 as compared to CRU estimates (Fig. 7). The only clear impact of surface albedo inaccuracy on annual average T2 is the relatively stronger cold bias in the Noah LSM (Fig. 7d). A closer inspection of T2 within Fig. 6 for the CRU dataset indicate that Noah-Sat, Noah, and Noah-MP all contain a domain-wide cold bias in annual average T2, while CLM-D and CLM-AF have minimal domain-wide T2 biases due to offsetting warm and cold biases in various regions. Several prior studies illustrate similar simulated T2 biases for African regions using WRF (e.g., Kerandi et

al., 2017; Li et al., 2015). The evaluation differences above indicate that the mean T2 bias/errors likely result from differences in the way radiative and surface energy fluxes are parameterized in the LSMs, since the patterns in T2 predictions do not follow differences in surface parameters and incoming solar radiation is roughly equivalent for all LSMs. This is further illustrated by the evaluation of T2MAX and T2MIN (Fig. S7). T2MAX is generally similar amongst all LSMs, except for Noah which contains a cold bias from the albedo overpredictions. The cold bias in Noah propagates to T2MIN, likely due to thermal inertia

from underestimated daytime heating. Both Noah-Sat and Noah-MP have various offsetting cold and warm T2MIN biases in the African-bioclimate regions, but CLM-D and CLM-AF both distinctly overpredict T2MIN. The overprediction of T2MIN in CLM-D and CLM-AF likely arises from the larger latent heat flux and upward surface long wave fluxes predicted by these LSMs (not shown), which may be related to the vegetation canopy approximation in CLM that does not account for gaps within the canopy or between vegetation crowns. These T2MIN overpredictions for CLM-D and CLM-AF also account for

the lack of annual average cold bias in these simulations. Additionally, the underpredicted T2MAX in the Noah LSM and overpredicted T2MIN in CLM-D and CLM-AF result in underpredicted DTR for these three LSMs.



The WRF comparison with the hourly NCDC-ISD dataset confirms the presence of a cold bias for the Noah LSM, but provides more insight into regional model performance. Across all the LSMs the wetter regions (e.g., MAD, WW, WWN, CW, LVW, EW) contain the strongest cold biases, while the semi-arid regions (e.g., SESD, WSD, NESD, SSD) contain the strongest warm

biases. This would appear to indicate that hourly temperature biases are modulated by inaccuracies in cloud radiative forcing or evaporative cooling.

The evaluation of the moisture variables PWV, CF, and PRE against MODIS and TRMM estimates (Fig. 8) and the spatial comparison of WRF PRE to observations (Fig. 9) show a more muted impact from LSM differences compared to temperature variables. Most regions have reasonable agreement in moisture variables with observations and satellite estimates, with a few

select regions experiencing very poor agreement (Fig 8). All LSM simulations overpredict PWV and CF, while underpredicting PRE. This indicates a possible underrepresentation of moisture recycling in this WRF configuration, whereby insufficient moisture convergence or insufficient activation of the cumulus parameterization fails to trigger precipitation, leading to excess water vapor that forms cloud cover. These findings are consistent with underpredictions in precipitation from the modified Tiedtke cumulus parametrization found by Igri et al. (2018), indicating that this cumulus scheme may be less efficient at

removing moisture from the atmosphere. The evaluation of Td2 and E2 against NCDC-ISD and CRU (supplementary Figs. S9 and S10) provide further evidence to support the possibility of insufficient moisture recycling as surface humidity is underpredicted, likely as a result of underpredicted PRE.

All LSM simulations reasonably capture the spatial pattern and magnitude of PRE compared to CRU, GPCP, and TRMM estimates (Fig. 9). Across all LSMs, PRE is better simulated in the wetter regions of West and Central Africa. The greatest

underpredictions occur in arid regions (ND, ED, SD, NESD, and WSD) and portions of East Africa (EM, CM, and LVW), while regions in South Africa (SSD and SM) and EW typically experience the strongest overprediction across the LSMs. Similar regional model biases have been reported in other studies (Alaka and Maloney, 2017; Argent et al., 2015; Cretat et al., 2012; Ratnam et al., 2018), indicating that our results are comparable to the large body of work utilizing WRF to study African precipitation. More details regarding moisture variable evaluation can be found in the Supplementary Material.

Lastly, comparisons of WSP10 to NCDC-ISD observations (Fig. S9 supplementary material) show a few key differences in WSP10 performance between LSMs. Noah and Noah-Sat have nearly identical overpredictions in the magnitude of WSP10, associated with an underestimation of RL. CLM-D also underpredicts the magnitude of WSP10, associated with the underrepresentation of RL in woody savannas and the inaccurate seasonal profile of RL. Both Noah-MP and CLM-AF have offsetting overpredictions and underpredictions in various regions, but both LSMs underpredict WSP10 in equatorial forested

areas, moderately underpredict or overpredict WSP10 in most moist vegetated regions, and largely overpredict WSP10 in more arid regions. The LSM regional model performance distribution may indicate that RL values in the forested regions are too large and the RL values in more semi-arid regions are too small in the Noah-MP and CLM-AF configurations.

Overall, the model validation experiments reveal little impact from inaccurate surface parameters on most meteorological parameters. The lack of poor meteorological performance may indicate that errors in surface parameters have minimal impacts





on African meteorology for certain applications. However, these errors can impact the trajectory of LULCC-induced climate
signals as demonstrated in Sect. 7.

## 7 Results - impact of LULCC on regional climate using different LSMs

Changes in land use and land cover, as represented by Dinamica Ego, between 2001 and 2015 are shown in Fig. 10. Broadly,
the LULC changes can be broken down into three categories: agricultural expansion, deforestation/degradation, and greening.

Agricultural expansion is defined here as the change in the LULC category from a natural vegetation type to either the MODIS
cropland or cropland/natural mosaic category. This LULCC is most prevalent across the northern and central portions of the
domain. In West Africa, a loss of evergreen broadleaf forest is found along the coasts of Ghana and Côte d'Ivoire, with woody
savanna significantly lost in Nigeria to cropland/natural vegetation mosaic. There are losses of savanna and grasslands to
cropland in Ethiopia, Sudan, and South Sudan, while losses of woody savanna to cropland/natural vegetation occur in the

western Republic of the Congo, western Democratic Republic of the Congo, and northwestern Angola.
Deforestation/degradation, defined here as the transition from a more forested MODIS natural vegetation type to a less forested
natural vegetation type, is commonly found in the southern and eastern portions of the domain.    Major
deforestation/degradation transitions include: a loss of woody savanna to savanna (e.g., central Angola, Mozambique, Zambia,
and Tanzania), loss of savanna to grasslands (e.g., Somalia and Kenya), and loss of savanna to open shrubland (e.g., Namibia,

Botswana, and Madagascar). Finally, greening, defined here as the reclamation of the barren MODIS category by a vegetated
category or a transition from a less forested vegetation category to a more forested vegetation category, is found along the
Saharan border, the boundary of the Namib Desert, within the Horn of Africa, and along the eastern coast of Madagascar.

### 7.1 LULCC impact on surface properties

A comparison of surface albedo changes between the LU01 and LUD simulations using the Noah, Noah-MP, CLM-D, and

CLM-AF LSMs is shown in Fig. 11. The CLM-AF LSM is consistent with expected changes.  Regions with a loss in vegetation
from either agricultural expansion or deforestation/degradation experience surface albedo increases, while areas with greening
experience albedo decreases. However, Noah, Noah-MP, and CLM-D all deviate from expected changes because of errors and
differences in their treatment of surface albedo. Additionally, due to the increased PFTs per LULC category in the CLM-AF
treatment there is greater overlap in PFTs between LULC categories, which results in albedo changes between vegetation

categories that are less extreme than the other LSMs.
The LULCC-induced albedo changes in Noah deviate the most from the other LSMs. This is largely because of the erroneous
treatment of woody savanna albedo as higher than croplands, cropland/natural vegetation mosaic, and savanna (Table S12).
The result of this flawed treatment is an erroneous albedo decrease in areas where woody savanna is lost to agricultural
expansion or deforested/degraded to savanna. While both CLM-D and Noah-MP also have inaccurate treatments for woody

savanna, these LSMs do not have erroneous albedo responses. For Noah-MP, this is because the savanna and cropland





categories are prescribed albedos less than woody savanna. In CLM-D, this is a result of the shrub leaf reflectance being less than that of grass and broad leaf deciduous trees. Noah-MP and CLM-D predict reductions in surface albedo for savanna to open shrubland transitions because both LSMs prescribe shrubs as having much lower leaf reflectance than grasses. In CLM-AF, the impact of lower shrub leaf reflectance is not as strong on the savanna to open shrubland transitions because open

shrublands contain more bare soil than savannas (Table 2), leading to albedo increases for savanna to open shrubland transitions. Noah-MP also does not show a change in albedo from the greening around the Sahara because its flawed soil color treatment does not simulate a significant difference in the albedo of grasslands and bare soil in that region (Fig. 2).

Among LSMs, there is greater similarity in LAI projections (Fig. 12) than for surface albedo (Fig. 11). The LAI projections from LULCC for CLM-AF and Noah-MP have the same spatial pattern and direction with slightly different magnitudes. The

projected LAI changes from CLM-D are also very similar to CLM-AF and Noah-MP across the northern half of the domain, but CLM-D has erroneous increases in LAI for woody savanna to savanna transitions. Again, these LAI errors are caused by erroneously treating woody savanna as a closed shrubland with a temporally uniform 1.0 $m^2$ $m^{-2}$ LAI (Table S2). The Noah LSM shows the greatest deviations from the other LSMs. This is mostly a result of erroneous increases in prescribed LAI values associated with agricultural expansion because croplands are prescribed higher LAI values than natural vegetation.

Additionally, the LAI of the woody savanna and savanna categories in Noah have the same prescribed values, hence this transition shows no change (see Table S14).

## 7.2 LULCC impact on 2 m temperature

Changes in T2 between the LU01 and LUD simulations for each LSM are shown in Fig. 13. Locations that have the largest magnitude differences in T2 align with the more localized changes in LAI and albedo. Similar T2 patters occur across the

northern half of the domain when comparing Noah-MP, CLM-D, and CLM-AF simulations, while Noah predicts the most unique changes. To further investigate the LULCC impacts, T2 differences are calculated for grid cells with different LULC transitions (see Table 10). Similarly, differences in the surface sensible and latent heat fluxes for these LULCCs are provided in supplementary tables S15 and S16.

Agricultural expansion induces annual average localized warming of ~0.1-0.2 °C using Noah-MP, CLM-D, and CLM-AF, but

a localized cooling of -0.12 °C using Noah because of its erroneous increase in LAI (Fig. 12) and subsequent enhanced evaporative cooling and cloud radiative forcing (not shown) in cropland transitions. The strongest warming response from agricultural expansion results from the loss of evergreen broad leaf forest to mosaic cropland along the coasts of Ghana and Côte d'Ivoire. This LULCC results in an average 0.6 °C warming using Noah-MP and ~1.3-1.4 °C of warming using CLM-D and CLM-AF, due to reductions in evapotranspiration (Table S16). Noah-MP predicts less warming amongst agricultural

expansion LULC transitions than CLM-AF. This is caused by the relative insensitivity of the surface latent heat fluxes (Table S16) to agricultural expansion, which allows for a loss in energy as the albedo increases (Fig. 11) along with other secondary feedbacks such as changes in cloud cover and precipitation discussed more below. CLM-AF consistently predicts reduced latent heat fluxes across the various agricultural expansion LULC transitions, resulting in the most overall warming. The





behaviour of CLM-AF is also consistent with the global remote sensing work of Duvellier et al. (2018), which indicates losses
in latent heat flux for all natural vegetation to cropland transitions. CLM-D has many T2 changes similar to CLM-AF with
some exceptions. For instance, the erroneous treatment of albedo for woody savanna in CLM-D, being too high, reduces the
surface sensible heat flux (Table S15) resulting in minor cooling.

Deforestation/degradation grid cells experience an average 0.22 °C warming using CLM-AF, while the remaining LSMs
predict almost no change in T2 for these grid cells (e.g., -0.03 – 0.04 °C). The strong warming signal in CLM-AF is again
related to larger evapotranspiration reductions with surface latent heat reductions exceeding 2 W m$^{-2}$ for all transitions (Table
S16), while Noah-MP has smaller changes in evaporation with surface latent heat reductions below 2 W m$^{-2}$ for all transitions.
CLM-D has the smallest overall change in T2 associated with deforestation/degradation due to offsetting changes in different
LULC transitions. This offsetting behaviour is primarily related to the woody savanna error that increases LAI and the latent
heat flux in woody savanna to savanna transitions. Since woody savanna to savanna transitions comprise a substantial portion
of the total deforestation/degradation grid cells, this signal cancels the warming from other transitions. Noah also experiences
offsetting impacts from different deforestation/degradation transitions; however, these changes are primarily driven by albedo
and changes in the surface sensible heat flux (Table S15). Noah predicts warming in the woody savanna to savanna transitions
because of significant erroneous albedo reductions, which causes substantial increases in the surface sensible heat flux. Noah
predicts cooling for the other dominant deforestation/degradation transitions due to albedo increases that are not sufficiently
compensated for by latent heat reductions. This behaviour indicates that the overpredicted albedo in Noah may substantially
shift the thermal sensitivity of LULCC to shortwave radiative effects rather than evapotranspiration effects.

Grid cells that experience greening have annual average cooling using Noah-MP, CLM-D, and CLM-AF. The strong cooling
in CLM-AF (-0.41 °C) and CLM-D (-0.33 °C) is the result of enhanced evapotranspiration, with CLM-D predicting less cooling
due to inaccurate surface property changes in savanna to woody savanna transitions. Noah-MP predicts much weaker cooling
(-0.13 °C) because vegetated to vegetated transitions experience little to no latent heat sensitivity, as discussed above. Finally,
the Noah simulations continue to be an outlier with almost no change (0.02 °C) due to offsetting inaccurate surface property
changes in different greening LULC transitions.

The three types of transition-based changes discussed above lead to very different spatial T2 changes (Fig. 13). The T2 changes
using the Noah LSM are largely incoherent due to various surface parameter errors. The T2 changes using Noah-MP are much
weaker than CLM-D or CLM-AF because only the starkest LULC transitions using Noah-MP impact local temperatures (i.e.,
transition from broad leaf evergreen forest to mosaic cropland within West Africa, transition from grassland to cropland in
northeastern Africa, and transition from barren soil to grassland along the Sahara border). The simulated T2 changes associated
with LULCC in CLM-D and CLM-AF are largely the same above the equator, but improper treatment of woody savannas and
southern hemisphere growing cycles result in erroneous cooling in southern Africa using CLM-D. CLM-AF is the only LSM
that captures warming from agricultural expansion in Nigeria, as well as the large-scale annual average warming associated
with deforestation/degradation in south-western Africa (e.g., Angola, Namibia, and Botswana).





### 7.3 LULCC impact on precipitation

In general, PRE changes between the LU01 and LUD simulations for each LSM (Fig. 14, Table 11) are more regional and much more chaotic than changes in temperature. However, there are a few localized changes in PRE from LULCC. Along the

coast of Ghana and Côte d'Ivoire, the lost broad leaf evergreen forest decreases PRE in all four LSMs by 0.12-0.45 mm day$^{-1}$ on average. This is in response to reduced moisture availability due to reduced evapotranspiration, enhanced stability from increased surface albedo, and possible reduced moisture convergence from reduced surface roughness. Additionally, both Noah-MP and CLM-D also predict reduced PRE for grid cells that experience woody savanna to mosaic cropland transitions (e.g., Nigeria), due to enhanced atmospheric stability from erroneous reductions in surface albedo.

The most significant regional PRE changes occur within southern Africa. During the southern Africa rainy season (October – March), the Angola Low is assumed to form in response to dry convection processes associated with surface heating in Angola (Mulenga 1998), however, the exact processes responsible for the Angola Low's formation are poorly understood (Munday and Washington, 2017). The strength and position of the Angola Low have been shown to significantly alter the gradients and magnitude of precipitation over southern Africa (e.g., Cook et al., 2004; Cretat et al., 2019). All LSMs predict excess heating

from deforestation/degradation between the LU01 and LUD simulations in Angola. This heating results in a persistent reduction of surface sea level pressure (Fig. S11), during southern Africa's rainy season (JFD), within Angola and nearby countries. The sea level pressure changes strengthens either the Angola Low or the associated Kalahari thermal low, which induces a stronger cyclonic circulation (Fig. S12) over southern Africa that opposes moist on-shore flow over Mozambique. This reduces moisture transport into south-western Africa, leads to drying in Angola and surrounding areas, and enhances

moisture convergence in south-eastern Africa increasing PRE in Mozambique and surrounding areas. The exact location and strength of this LULCC-induced PRE climate signal varies between LSMs due to differences in the strength and spatial location of maximum heating, but this feature appears robust.

### 8 Summary and conclusions

In this work the applicability of commonly used WRF LSMs (i.e., Noah, Noah-MP, and CLM-D) with WRF's default MODIS

LULC data are explored in Sub-Saharan Africa. Each default WRF LSM is found to have unique deficiencies in representing African surface parameters including: 1) significantly overestimated surface albedo and underestimated surface roughness using the Noah LSM, 2) the same underestimated surface roughness as Noah using Noah-Sat, 3) significantly underestimated surface albedo in arid areas due to inaccurate soil albedo treatments using Noah-MP, and 4) geographically invariable surface parameters using CLM-D that make it unsuitable for use outside the Northern Hemisphere Mid-Latitudes. Additionally, all

default WRF LSMs inaccurately treat the MODIS woody savanna land use category as closed shrubland. These deficiencies likely have a minimal impact on simulations in middle or high latitudes of the Northern Hemisphere, but lead to substantial inaccuracies in Africa. Consequently, we developed a version of the CLM LSM in WRF that more accurately represents these properties in Africa (CLM-AF).





Despite clear deficiencies in surface parameters, all WRF LSMs reasonably capture the spatial pattern and magnitudes of
precipitation and T2. The only detectable impact of inaccurate surface parameters is the slightly stronger cold and dry bias
using the Noah LSM that occurs because of its overestimated albedo. The WRF model with each LSM reasonably captures
the climate of Sub-Saharan Africa, despite errors with cloud parameters and radiative forcing that are common to most climate
models (e.g., Lauer and Hamilton 2013).

Regardless of the similar meteorological performance, the land surface parameter errors amongst the default WRF LSMs
substantially impact the magnitude and direction of LULCC-induced changes in temperature and to a lesser extent localized
changes in precipitation. The surface parameters in the Noah LSM and CLM-D are the most flawed, and as a result neither
LSM is suitable for LULCC experiments in Africa. Additionally, great care should be taken when utilizing these LSMs for
other scientific applications in these regions. Noah-MP is least flawed of the default LSMs and with several updates may also
be suitable for use in tropical regions (e.g., Spera et al., 2018).

Although several of the default LSMs produced erroneous LULCC-induced climate signals, there are several common features
that stand out as potentially robust. Losses of broad leaf evergreen forest along the coasts of Ghana and Côte d'Ivoire to
agricultural expansion between 2001 and 2015 appear to have caused warming and drying in this region for LSMs that
accurately treat this transition. Additionally, warming from deforestation in Angola, Namibia, and Botswana are modelled to
have altered the JFD average atmospheric circulations in this region, decreasing precipitation in south-western Africa and
increasing precipitation in south-eastern Africa. Important mechanistic differences also stand out between the Noah-MP and
CLM-AF LSMs. Noah-MP predicts little change in latent heat flux between vegetated to vegetated LULC transitions unless
they are particularly stark (e.g., broad leaf evergreen forest to mosaic cropland), while CLM-AF consistently predicts latent
heat flux change between vegetation transitions resulting in stronger thermal changes from gains or losses in evaporative
cooling. This indicates that the accuracy of the latent heat sensitivity of LSMs to LULCC is crucial to the accuracy of LULCC
climate signal predictions in the tropics. Additionally, the incoherent temperature and moisture climate signals in the Noah
LSM indicate that albedo accuracy may play a role in determining whether evapotranspiration or shortwave radiative effects
will dominate LULCC climate signals.

Overall, this study serves as a cautionary tale to illustrate that proficient meteorological performance can mask severe flaws in
model treatments, and that special care is needed to evaluate LSM parameters when conducting LULCC studies in Africa.
While this study focuses on Africa, we expect that these LSMs would encounter similar problems in applications to other
regions of the tropics or Southern Hemisphere. More work is required by the scientific and model development communities
to not only improve meteorological model processes, but to ensure that these scientific improvements are applicable to as many
climate regimes and localities as possible. Additionally, this work documents the development of the WRF CLM-AF
configuration for use in LULCC studies of Sub-Saharan Africa. Future companion manuscripts will explore the climate change
signals attributable to LULCC in Sub-Saharan Africa, their statistical significance, and their impact on air quality. This
development is a first step towards better global LULC representations in WRF, but additional improvements are needed to
accurately represent land surface and vegetation parameters across the various global climate regimes.

*Code and data availability.* The default WRF model is publically available for download from the WRF website
(https://www2.mmm.ucar.edu/wrf/users/downloads.html). The CLM-AF code, additional code to recreate the experiments
shown here, the African bio-climate region data, the Dinamica EGO generated land use and land cover data, and instructions
for using these codes and input data are all available on the UNC Dataverse Archive (https://dataverse.unc.edu/dataverse/CLM-
AF1). The ERA-Interim reanalysis data for meteorological initial and boundary conditions can be found on the NCAR
Research Data Archive website (https://rda.ucar.edu/datasets/ds627.0/). All observational data used to evaluate the WRF
model are publically available from the websites listed in Table 8.

*Author contributions.* Timothy Glotfelty developed and tested the CLM-AF code, conducted all WRF simulations, and wrote
the manuscript. Jared Bowden and J. Jason West provided scientific input on the code development and experimental design.
Jared Bowden processed and refined the manuscript figures. Diana Ramirez and Adrian Ghilardi designed the land use
modelling framework. Diana Ramirez generated all the land use and land cover modelling products. All authors reviewed and
modified the manuscript.

*Competing interests.* The authors declare that they have no conflict of interest

*Acknowledgements.* Special thanks are due to Tanya Spero, Jonathan Pleim, and Limei Ran of the United States Environmental
Protection Agency for their valuable feedback on the development of CLM-AF. This work is sponsored by the National Science
Foundation Coupled Natural Human Systems program grant 1617359 and the National Institute for Environmental Health
Sciences institutional training grant T32ES007018. Computational resources were provided by Extreme Science and
Engineering Discovery Environment (XSEDE), which is supported by National Science Foundation grant number ACI-
1548562. Simulations were conducted on the XSEDE stampede2 cluster provided by the Texas Advanced Computing Center,
through allocation TG-ATM180014.

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

# African Bioclimate Regions

| | | |
|---|---|---|
| 1 North Dry | 6 West Semi-Dry | 11 Central Wet (S) | 16 Lake Victoria Wet | 21 South Semi-Dry |
| 2 East Dry | 7 East Wet | 12 Central Wet (SA) | 17 East Moist | 22 South Moist |
| 3 Northeast Semi-Dry (N) | 8 West Moist | 13 West Wet Nigeria | 18 Southeast Semi-Dry | |
| 4 Northeast Semi-Dry (S) | 9 West Wet | 14 Central Moist (N) | 19 Madagascar | |
| 5 Northeast Semi-Dry (SH) | 10 Central Wet (N) | 15 Central Moist (S) | 20 South Dry | |

**Figure 1: African bioclimate and sub-bioclimate regions defined in this study within the Sub-Saharan domain.**

**Figure 2: Comparison of annual average albedo (%) between LSM configurations. Since Noah-Sat is based on satellite observations, it can be treated as observations.**



**Figure 3: Comparison of annual, summer (JJA), and winter (JFD) average LAI (m² m⁻²) between LSM configurations. Since Noah-Sat is based on satellite observations, it can be treated as observations.**



**Figure 4: Comparison of annual average surface roughness length (m) between LSM configurations.**





**Figure 5: 2013 annual average upwelling shortwave radiation at the Earth's surface (W m$^{-2}$) for CERES-EBAF estimates and WRF simulations.**



Figure 6: Normalized mean error and mean bias statistics of USRS and T2 relative to observations, CERES–EBAF USRS estimates and CRU/NCDC T2.








**Figure 7: 2013 annual average 2 m temperature (°C) from CRU and WRF simulations with five different LSMs. NCDC-ISD 2 m temperature observations are overlaid.**

**Figure 8: Normalized mean error and mean bias statistics of PWV, cloud fraction, and precipitation relative to observed MODIS PWV/cloud fraction and TRMM precipitation.**


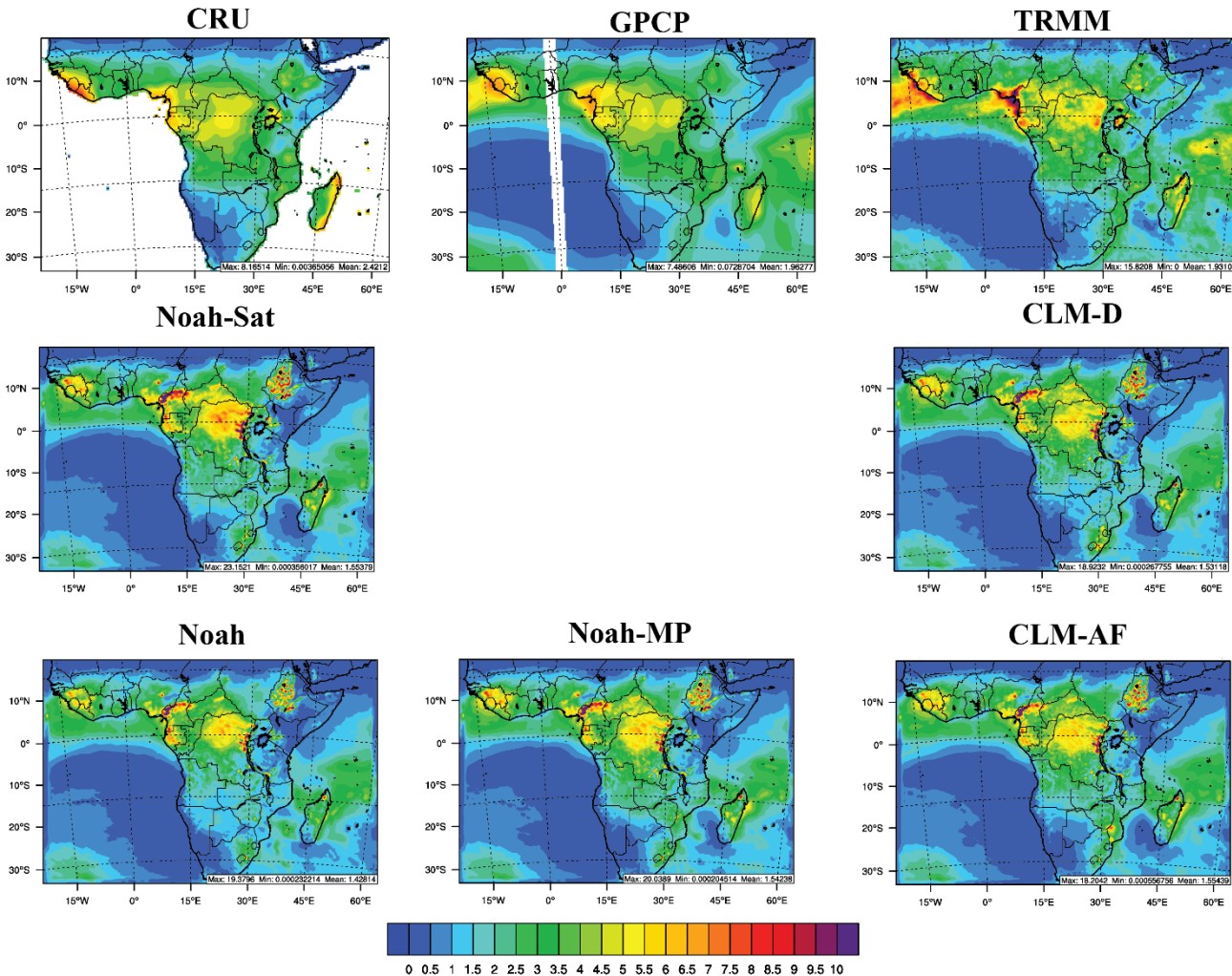

**Figure 9: 2013 annual average precipitation rate (mm day⁻¹) from CRU, GPCP, TRMM, and the five WRF LSM simulations.**






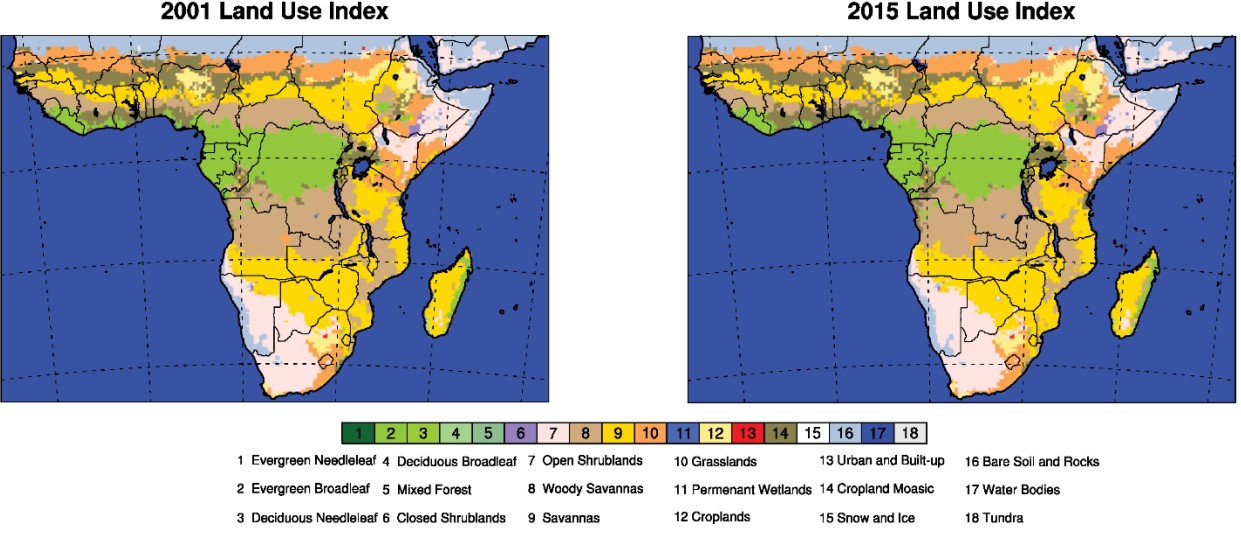

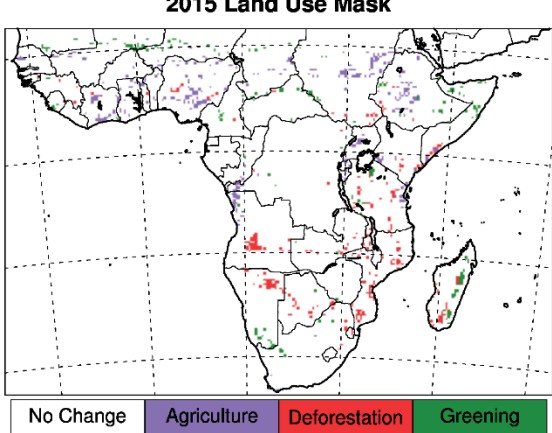

**Figure 10: Processed MODIS land use and land cover categories for 2001, simulated categories for 2015, and grid cells that**
**experience transitions due to agricultural expansion, deforestation/degradation, and greening.**




**Figure 11: Differences in albedo (%) between LUD and LU01 simulations using Noah, Noah-MP, CLM-D, and CLM-AF.**


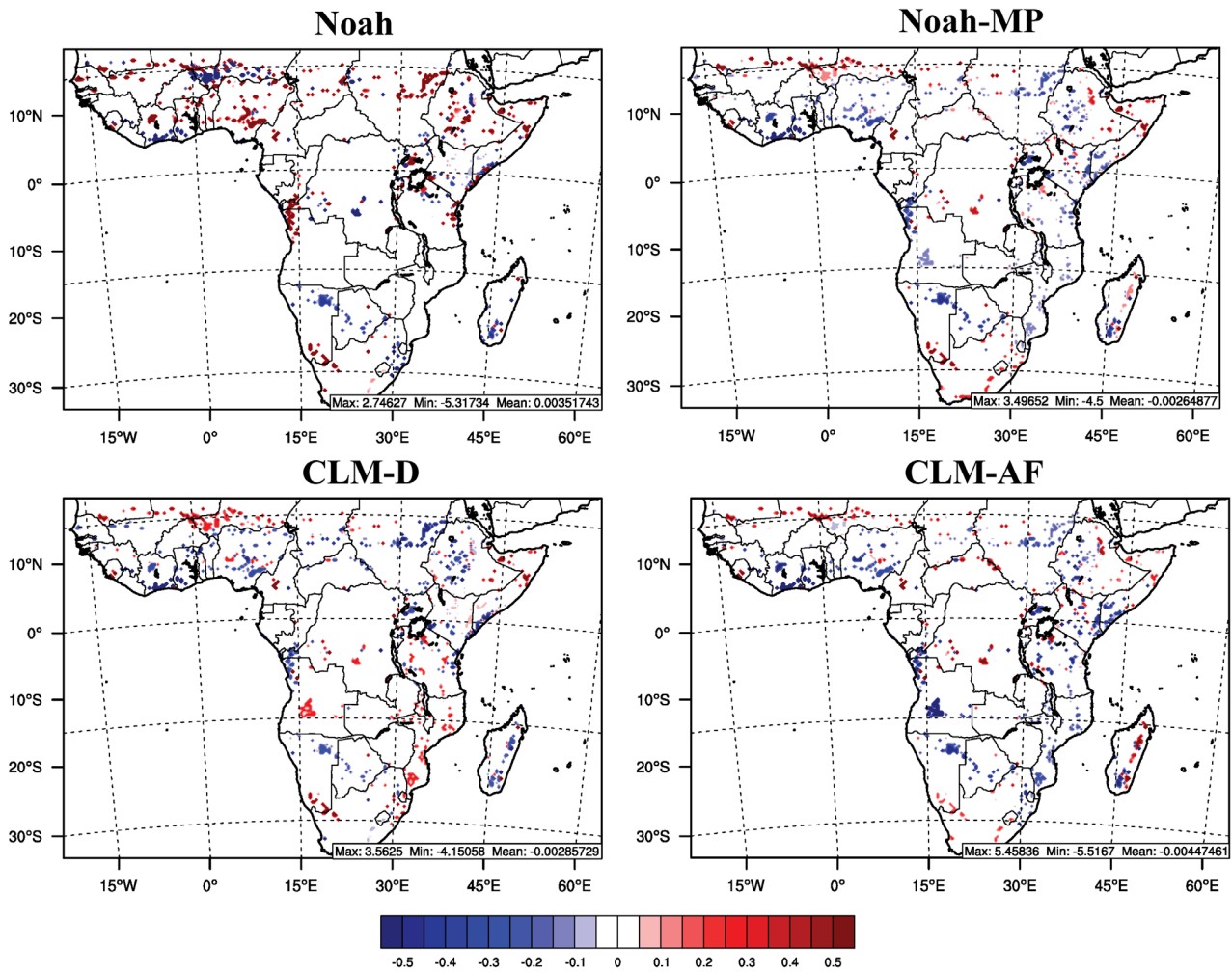

**Figure 12: Differences in leaf area index (m² m⁻²) between the LUD and LU01 simulations using Noah, Noah-MP, CLM-D, and CLM-AF.**


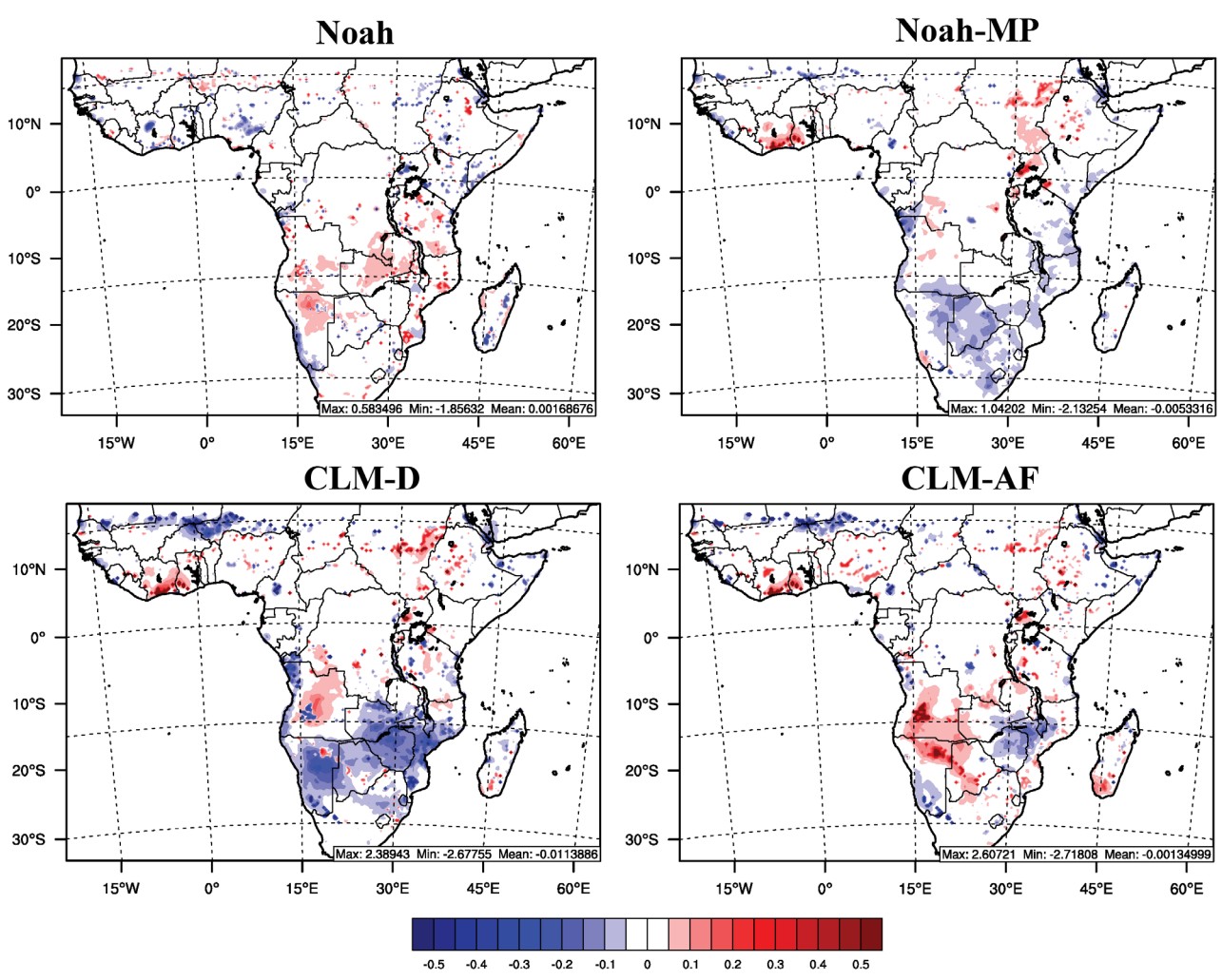


**Figure 13: Differences in 2 m Temperature (°C) between the LUD and LU01 simulations using Noah, Noah-MP, CLM-D, and CLM-AF.**


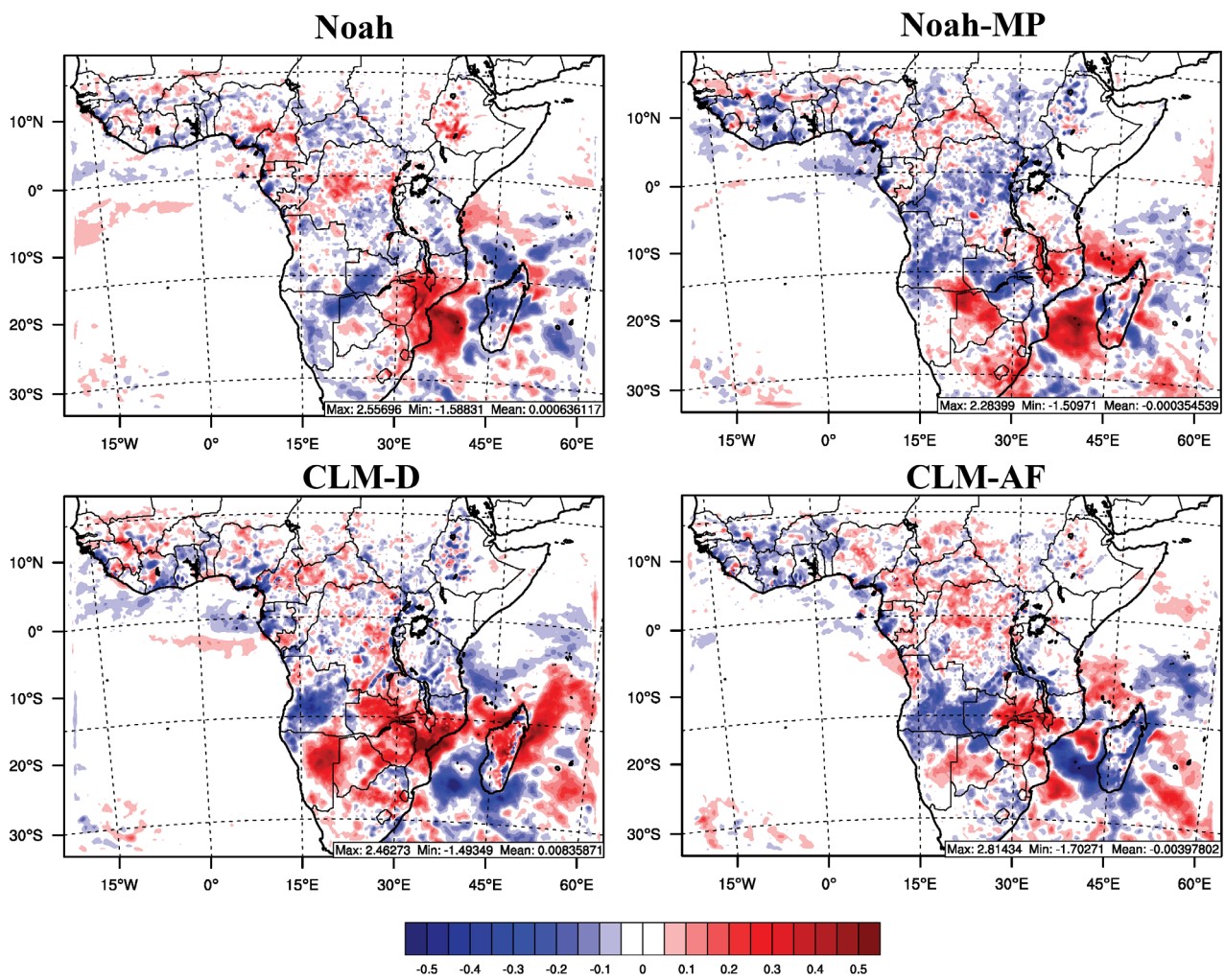

**Figure 14: Differences in precipitation rate (mm day⁻¹) between the LUD and LU01 simulations using Noah, Noah-MP, CLM-D, and CLM-AF.**



Table 1: Model Configurations

| Simulation Configuration | Setting |
| --- | --- |
| Domain | Sub-Saharan Africa |
| Horizontal Resolution | 36 km |
| Vertical Layers | 30 Layers from the Surface to 50 mb |
| Initial/Boundary Conditions | ERA-Interim (D11) |
| **Physics Parameterization** | **Option** |
| Cumulus | New Tiedtke Scheme (Z11) |
| Cloud Microphysics | Aerosol-Aware Thompson Scheme (TE14) |
| Radiation | RRTMG (C05; I08) |
| Planetary Boundary Layer | MYNN (NN04; NN06) |
| Surface Layer | MYNN (NN04; NN06) |
| Land Surface Models | Noah (CD01; E03) |
|  | Noah-MP (N11) |
|  | CLM 4.5 (S11; JW12; LK12) |
| Lake Model | CLM 4.5 (S12; G13) |

Acronyms: ERA-Interim – European Centre for Medium-Range Weather Forecasting Interim
reanalysis; RRTMG – Rapid Radiative Transfer Model for General Circulation Models; MYNN –
Mellor Yamada Nakanishi Niino; Noah - National Centers for Environmental Prediction, Oregon State
University, Air Force, Hydrology Lab; Noah-MP – Noah Multi-patameterization options; CLM 4.5 –
Community Land Model version 4.5

References: D11 – Dee et al., (2011); Z11 – Zhang et al., (2011); TE14 – Thompson and Eidhammer,
        (2014); C05- Clough et al., (2005); I08 – Iacono et al., (2008); NN04 – Nakanishi and Niino,
        (2004); NN06 – Nakanishi and Niino, (2006);  CD01 – Chen and Dudhia (2001); E03 – Ek et
al., (2003); N11 – Niu et al., (2011); S11 – Subin et al., (2011); JW12 – Jin and Wen, (2012);
        LK12 – Lu and Kueppers, (2012); S12 – Subin et al., (2012); G13 – Gu et al., (2013)








Table 2: Percentage of Plant Functional Types Assigned to MODIS Land Use Categories in the Updated CLM-AF

| MODIS Land Use Category | 2 | 4 | 5 | 6 | 7 | 8 | 9 | 10 | 12 | 14 |
|---|---|---|---|---|---|---|---|---|---|---|
| Bare Soil | - | 3 | - | 14 | 48 | - | - | 21 | 10 | - |
| Broad Leaf Evergreen Tropical Tree | 82 | - | 20 | - | - | 12 | - | - | - | - |
| Broad Leaf Evergreen Temperate Tree | 18 | - | 15 | - | - | - | - | - | - | - |
| Broad Leaf Deciduous Tropical Tree | - | 55 | - | 8 | - | 26 | 21 | - | - | 24 |
| Broad Leaf Deciduous Temperate Shrub | - | 18 | - | 57 | 31 | - | - | 18 | - | - |
| C3 Non-Artic Grass | - | - | 40 | - | 8 | 27 | 31 | 36 | 24 | 17 |
| C4 Grass | - | 24 | 25 | 21 | 13 | 35 | 48 | 25 | 15 | 33 |
| Corn | - | - | - | - | - | - | - | - | 51 | 26 |

MODIS Land Use Categories: 2 – Evergreen Broad Leaf Forest; 4 – Deciduous Broad Leaf Forest; 5 – Mixed Forest; 6 – Closed Shrublands; 7 – Open Shrublands; 8 – Woody Savanna; 9 – Savannas; 10 – 
Grasslands; 12 – Croplands; 14 –Cropland/Natural Mosaic

Table 3: Dominant MODIS-IGBP Land Use Categories within African Bioclimate Regions at 36 km Resolution

| Region | | MODIS-IGBP Category | | | | | | | | | |
|---|---|---|---|---|---|---|---|---|---|---|---|
| Name | Acronym | 2 | 4 | 5 | 6 | 7 | 8 | 9 | 10 | 12 | 14 |
| North Dry | ND | - | - | - | - | Y | - | - | Y | Y | Y |
| East Dry | ED | - | Y | - | - | Y | - | - | Y | - | Y |
| Northeast Semi-Dry[*] | NESD | - | - | - | Y | Y | Y | Y | Y | Y | Y |
| West Semi-Dry | WSD | - | - | - | - | Y | - | Y | Y | Y | Y |
| East Wet | EW | Y | - | - | - | - | Y | Y | Y | Y | Y |
| West Moist | WM | - | - | - | - | - | Y | Y | - | Y | Y |
| West Wet | WW | Y | - | - | - | - | Y | Y | Y | - | Y |
| Central Wet[*] | CW | Y | - | - | - | Y | Y | Y | Y | - | Y |
| West Wet Nigeria | WWN | Y | - | - | - | - | Y | Y | - | Y | Y |
| Central Moist[*] | CM | Y | - | - | - | - | Y | Y | Y | Y | Y |
| Lake Victoria Wet | LVW | Y | - | - | - | - | Y | Y | Y | Y | Y |
| East Moist | EM | Y | - | Y | - | - | Y | Y | Y | - | Y |
| Southeast Semi-Dry | SESD | - | - | - | - | Y | Y | Y | - | - | - |
| Madagascar | MAD | Y | Y | - | - | Y | Y | Y | Y | - | Y |
| South Dry | SD | - | - | - | - | Y | - | - | - | - | - |
| South Semi-Dry | SSD | Y | - | - | Y | Y | Y | Y | Y | Y | - |
| South Moist | SM | Y | - | - | - | Y | Y | Y | Y | Y | Y |

[*]: Indicates bioclimate regions that are subdivided into a north, a south, or other sub-bioclimate regions for better LAI geographical distributions.






Table 4: Regional Interpolation of Missing PFT data

| Region | First Region | PFTs | Second Region | PFTs |
|--------|--------------|------|---------------|------|
| ND | WW | 4,5 | WSD | 6,10,11 |
| ED | EW | 4,5 | NESD-N | 13,14,15 |
| NESD-N | EW | 4,5 | - | - |
| NESD-S | EW | 4,5 | - | - |
| NESD-SH | EW | 4,5 | NESD-S | 6,15 |
| WSD | WW | 4,5 | - | - |
| EW | NESD-N | 10 | - | - |
| WM | WW | 4,5 | WSD | 10,13 |
| WW | WSD | 10 | - | - |
| CW-N | WSD | 10 | - | - |
| CW-S | SESD | 15 | - | - |
| CW-SA | CW-S | 6 | SESD | 15 |
| WWN | WW | 6 | WSD | 10 |
| CM-N | CW-N | 4,5,6 | NESD-S | 10 |
| CM-S | CW-S | 6,10 | - | - |
| LVW | NESD-S | 6,10 | - | - |
| EM | CW-S | 6 | SESD | 10 |
| SESD | EM | 4,5 | - | - |
| MAD | SESD | 15 | - | - |
| SD | SSD | 4,5,6,14,15 | - | - |
| SSD | - | - | - | - |
| SM | SSD | 14 | - | - |

Table 5: Equations for Calculating SAI for Each PFT in CLM-AF

| PFT | SAI Equation | SAI Minimum |
|-----|--------------|-------------|
| Broad Leaf Evergreen Trees | $SAI = -\Delta LAI + 0.5$ | 0.5 |
| Broad Leaf Deciduous Tropical Trees | $SAI = -1.0385(\Delta LAI) + 0.2$ | 0.3 |
| Broad Leaf Deciduous Shrubs | $SAI = -0.8(\Delta LAI) + 0.12$ | 0.1 |
| C3 Non-Arctic Grass | $SAI = -0.9(\Delta LAI) + 0.32$ | 0.1 |
| C4 Grass and Corn | $SAI = -\Delta LAI + 0.3$ | 0.3 |

SAI: Stem Area Index; $\Delta LAI$: Difference between the LAI of the current and previous month




Table 6: Sandy Soil CLM Albedo Values

| Moisture | Radiation Band | CLM-D | | CLM-AF | |
|---|---|---|---|---|---|
| | | Sand | Sand-Loam | Sand | Sand-Loam |
| Saturated | Visible | 0.12 | 0.11 | 0.22 | 0.13 |
| | Infrared | 0.24 | 0.22 | 0.34 | 0.24 |
| Dry | Visible | 0.24 | 0.22 | 0.34 | 0.24 |
| | Infrared | 0.48 | 0.44 | 0.58 | 0.46 |



Table 7: Model Experiments and Simulations

| Experiment | Period | Land Use | LSM |
|---|---|---|---|
| Model Validation | 2013 | Default | Noah |
| | | | Noah-Sat |
| | | | Noah-MP |
| | | | CLM-D |
| | | | CLM-AF |
| Land Use Land Cover Change | 2010-2015 | MODIS 2001 (LU01) | Noah |
| | | | Noah-MP |
| | | | CLM-D |
| | | | CLM-AF |
| | | Dinamica EGO 2010-2015 (LUD) | Noah |
| | | | Noah-MP |
| | | | CLM-D |
| | | | CLM-AF |







Table 8: Datasets for Model Validation

| Datasets | Temporal Resolution | Spatial Resolution | Website |
|---|---|---|---|
| CRU TS4.02[a] | Monthly | 0.5º×0.5º | https://crudata.uea.ac.uk/cru/data/hrg/cru_ts_4.02/ |
| NCDC-ISD[b] | Hourly | n/a | https://www.ncdc.noaa.gov/land-based-station-data/integrated-surface-database-isd |
| CERES-EBAF[c] | Monthly | 1º×1º | https://ceres.larc.nasa.gov/ |
| GPCP[d] | Monthly | 2.5º×2.5º | http://www.esrl.noaa.gov/psd/data/gridded/data.gpcp.html |
| TRMM[e] | 3-Hour | 0.25º×0.25º | https://pmm.nasa.gov/data-access/downloads/trmm |
| MOD08_M3[f] | Monthly | 1º×1º | https://modaps.modaps.eosdis.nasa.gov/services/about/products/c61/MOD08_M3.html |

[a]: University of East Anglia, Climate Research Gridded Climate Data version 4.02; [b]: National Climate
Data Center – Integrated Surface Data; [c]: Clouds and Earth's Radiant Energy System – Energy Balanced
and Filled; [d]: Global Precipitation Climatology Project; [e]: Tropical Rainfall Measuring Mission; [f]:
MODIS/Terra Aerosol Cloud Water Vapor Ozone Monthly L3 Global 1Deg CMG


Table 9: Evaluated Variables and Evaluation Datasets

| Variable | Acronym | Evaluation Dataset |
|---|---|---|
| 2 m Temperature | T2 | CRU TS4.02 and NCDC-ISD |
| Daily Maximum Temperature | TMAX | CRU TS4.02 |
| Daily Minimum Temperature | TMIN | CRU TS4.02 |
| Diurnal Temperature Range | DTR | CRU TS4.02 |
| 2m Vapor Pressure | E2 | CRU TS4.02 |
| 2 m Dew point Temperature | $T_d2$ | NCDC-ISD |
| Precipitable Water Vapor | PWV | MOD08_M3 |
| Cloud Fraction | CF | CRU TS4.02 and MOD08_M3 |
| Precipitation | PRE | CRU TS4.02, GPCP, and TRMM |
| 10 m Wind Speed | WS10 | NCDC-ISD |
| Dowelling Shortwave Radiation (Surface) | DSR | CERES-EBAF |
| Dowelling Longwave Radiation (Surface) | DLR | CERES-EBAF |
| Upwelling Shortwave Radiation (TOA[*]) | USRT | CERES-EBAF |
| Upwelling Shortwave Radiation (Surface) | USRS | CERES-EBAF |
| Upwelling Longwave Radiation (TOA[*]) | ULR | CERES-EBAF |
| Shortwave Cloud Forcing | SWCF | CERES-EBAF |
| Longwave Cloud Forcing | LWCF | CERES-EBAF |

[*]: Top of the Atmosphere







Table 10: Annual Average 2 m Temperature Change (°C) in WRF Grid Cells that experience LULCCs
between 2001 and 2010-2015

| Transition | Noah | Noah-MP | CLM-D | CLM-AF |
|---|---|---|---|---|
| **Agricultural Expansion**[*] | -0.12 | 0.1 | 0.1 | 0.17 |
| 10 to 12 | -0.09 | 0.16 | 0.18 | 0.17 |
| 2 to 14 | -0.3 | 0.6 | 1.34 | 1.38 |
| 8 to 14 | -0.06 | 0.01 | -0.12 | 0.15 |
| 10 to 14 | -0.1 | 0.06 | 0.03 | 0.07 |
| **Deforestation/Degradation**[*] | 0.04 | -0.01 | -0.03 | 0.22 |
| 8 to 9 | 0.17 | -0.03 | -0.22 | 0.18 |
| 9 to 7 | -0.16 | -0.04 | 0.12 | 0.36 |
| 9 to 10 | -0.11 | -0.05 | 0.1 | 0.11 |
| **Greening**[*] | 0.02 | -0.13 | -0.33 | -0.41 |
| 9 to 8 | -0.12 | 0.0 | 0.08 | -0.13 |
| 10 to 9 | 0.18 | -0.02 | -0.28 | -0.26 |
| 16 to 7 | -0.01 | -0.13 | -0.39 | -0.40 |
| 16 to 10 | 0.09 | -0.2 | -0.81 | -0.8 |

[*]: Shows average difference for a broad class of LULCC followed by the average difference in the
major MODIS LULC transitions that comprise that class.  MODIS Land Use Categories: 2 – Evergreen
Broad Leaf Forest; 7 – Open Shrublands; 8 – Woody Savanna; 9 – Savannas; 10 – Grasslands; 12 –
Croplands; 14 –Cropland/Natural Mosaic; 16 – Barren/ Sparsely Vegetated








Table 11: Annual Average Precipitation Rate Change (mm day$^{-1}$) in WRF Grid Cells that experience LULCCs between 2001 and 2015

| Transition | Noah | Noah-MP | CLM-D | CLM-AF |
|---|---|---|---|---|
| **Agricultural Expansion**[*] | 0.02 | -0.13 | -0.08 | -0.03 |
| 10 to 12 | 0.02 | -0.04 | 0.05 | -0.02 |
| 2 to 14 | -0.12 | -0.25 | -0.45 | -0.38 |
| 8 to 14 | 0.07 | -0.18 | -0.10 | 0.00 |
| 10 to 14 | 0.04 | -0.01 | 0.04 | -0.02 |
| **Deforestation/Degradation**[*] | 0.02 | -0.01 | -0.04 | -0.08 |
| 8 to 9 | 0.07 | -0.03 | -0.01 | -0.04 |
| 9 to 7 | -0.05 | 0.05 | 0.12 | -0.05 |
| 9 to 10 | -0.01 | -0.01 | -0.02 | -0.04 |
| **Greening**[*] | 0.00 | 0.03 | 0.15 | 0.05 |
| 9 to 8 | -0.02 | 0.03 | 0.14 | -0.03 |
| 10 to 9 | -0.03 | 0.06 | 0.01 | 0.02 |
| 16 to 7 | -0.01 | 0.02 | 0.05 | 0.00 |
| 16 to 10 | 0.05 | -0.01 | 0.02 | 0.02 |

[*]: Shows average difference for a broad class of LULCC followed by the average difference in the major MODIS LULC transitions that comprise that class. MODIS Land Use Categories: 2 – Evergreen
Broad Leaf Forest; 7 – Open Shrublands; 8 – Woody Savanna; 9 – Savannas; 10 – Grasslands; 12 – Croplands; 14 –Cropland/Natural Mosaic; 16 – Barren/ Sparsely Vegetated