# Peer review of "Limitations of WRF land surface models for simulating land use and land cover change in Sub-Saharan Africa and development of an improved model (CLM-AF v. 1.0)"

_Geoscientific Model Development, 2020_

## Referee Comment (RC1) · Anonymous Referee #1 · 10 Nov 2020

Glotfelty et al. test the applicability of different LSMs incorporated in WRF RCM for Sub-Saharan Africa and their suitability to simulate the effects of land use changes on the regional climate conditions adequately. The results show that surface albedo, leaf area index and surface roughness are not accurately represented in the default models. Therefore, a new version of WRF coupled to the CLM LSM is developed, adjusted to the specific surface conditions in Sub-Saharan Africa.

The topic of the study is within the scope of GMD and relevant for the large WRF modeling community in Africa and even beyond in the context of land use change effects on the regional climate in Africa. The manuscript is well structured and comprehensively written. The motivation of the paper is clear and the methods are well documented. Nevertheless, I have some concerns regarding the experimental setup, which need to be addressed by the authors before the manuscript is suitable for publication.

**Major comments:**

1) In a first step, a model validation experiment is performed, in which simulations with five different models are conducted for one year. In general, I would say one year simulations are rather short to validate model performances, especially when only annual averages are presented. The authors argue that they chose the year 2013 because its a neutral year for the El Nino Southern Oscilation, but they do not really explain why this is the ideal boundary condition to validate the LSM performances. In any case, it is a single year that cannot represent the whole climate variability. Deviating atmospheric circulation conditions can considerably affect the impact of the land surface conditions on the regional climate conditions. Moreover, due to the exclusive consideration of annual averages, seasonal conditions are excluded in which the land surface conditions have larger impacts on the regional climate (e.g. dry conditions). Therefore, I recommend to extend the simulation period, or at least, to consider seasonal effects.

In a second step, it is intended to quantify the impact of land use changes on the simulation results with different LSMs. For this, the results of climate simulations for the period 2001-2010 with static land use conditions are compared to results of climate simulations for the period 2010-2015, including observed land use changes. But differences between two simulations with different land use conditions do not have to be caused inevitably/exclusively by the different land use conditions, in the case of deviating simulation periods. The different atmospheric circulation conditions in both periods can have certain impacts on the simulation results. Thus, from my point of view, identical simulation periods would have been preferable (2001-2015). If it is not possible to perform these simulations with respect to computing time, one could eventually reduce the number of LSMs based on the results of the validation experiment. By the way, I do not really understand why Noah Sat is included in the study, if one cannot use it for land use change scenarios at all. The authors should at least discuss the potential effects of the different simulation periods.

2) It is very difficult to assess the differences between the different LSMs in the validation experiment based on the shown figures. It is therefore very difficult to compare these differences to the changes caused by land use changes. Plots of the differences to observations as shown in section 7 would help a lot.

3) To be able to understand the results of the validation experiment comprehensively, an assessment of the sensible and latent heat fluxes is necessary.

**Minor comments:**

1) The biogeophysical effects of the surface roughness on the climate impacts of land use changes is not considered. For instance, this impact can be seen for the deforestation regions. The model results consistently show a warming with deforestation. In the manuscript, this is explained by

reduced evapotranspiration rates and an associated reduced latent cooling. But if the reduced latent heat fluxes are the reason for the increased near-surface temperatures, accordingly the sensible heat fluxes should be increased (due to the increased temperature gradient between the land surface and the atmosphere). But this is not the case, the sensible heat fluxes are also reduced. Therefore, I suppose that the efficiency of the deforested land surface to transform the incoming solar energy in turbulent heat is reduced due to the reduced surface roughness, resulting in a warming of the surface (e.g. Winckler et al., 2019; Breil et al., 2020).

2) simulation results show that the cloud cover is consistently overestimated in the validation experiment. At the same time, downward short-wave radiation (swdown?) is also overestimated. How does that fit together?

3) Several abbreviations are used which are not explained in the text (e.g. SWDOWN, GLW, OLR, SWUPT).

**References:**

Breil, M., and Coauthors, (2020). The Opposing Effects of Reforestation and Afforestation on the Diurnal Temperature Cycle at the Surface and in the Lowest Atmospheric Model Level in the European Summer. *Journal of Climate*, *33*(21), 9159-9179.

Winckler, J., and Coauthors, 2019: Different response of surface temperature and air temperature to deforestation in climate models. Earth Syst. Dyn., 10, 473–484, https://doi.org/10.5194/esd-10-473-2019.

---

## Referee Comment (RC2) · Anonymous Referee #2 · 10 Feb 2021

In this study, Glotfelty et al. assess the ability of several land surface models to simulate the surface characteristics of sub-Saharan Africa, within the WRF regional numerical weather prediction framework. They find that the default models do a rather poor job in the region, in terms of, for instance, albedo and leaf area index. Consequently, they develop a new CLM land surface model variant, wherein they have improved the representation of such surface properties. Following, they perform a land use/land cover change experiment to highlight the applicability of the new variant, since they show how the meteorological proficiency of the default models hides problems in surface processes, which can lead to flawed conclusions in such experiments. I agree with RC1 that the topic is within the scope of GMD and that this work is relevant not only for the WRF modelling community, but also for land use/land cover change studies focusing on Africa and the Southern Hemisphere. I also find the manuscript to be well outlined, with a clear motivation, sufficient methods, and an in-depth analysis. However, I think it could be improved by addressing the following specific comments. Additionally, I list some technical details that I think should be corrected.

**Specific comments**

1. After reading the major comment 1) in RC1, it is clear to me that authors need to improve a few things in Section 4 to better explain their experimental setup. I agree in Section 4.1 they should address RC1 comments, and justify why a single-year validation is enough for the scope of the study, and why it is desirable it is not during an El Niño–Southern Oscillation phase. Maybe a reference to a similar study could help, or stating explicitly the proof-of-concept nature of the validation. Perhaps avoid use the word "validation", in case other studies have validated these LSMs in Africa.

2. Then I think Sections 4.2 and 4.3 are failing to explain the time period simulated. The comment in RC1 confuses me, because what I understand is that simulations are for boundary conditions of period 2010–2015, all of them, only that in LU01 land use maps are from MODIS 2001, and in LUD land use maps are from Dinamica EGO results. Is this right? If so, then comparison between LU01 and LUD is done subtracting 6-year (2010–2015) averages. Is this right? Authors should explain why this time span was chosen. I think the different time periods in Section 4.3, to explain the Dinamica EGO methods, may confuse the reader. Why mention year 2050? Why add in line 317: "and to 2050 for simulation purposes"?.

3. Minor comment 2) of RC1: does it perhaps have to do with the fact that SWDOWN

is compared with CERES and CF with MODIS?

4. I think also a table with the data used from ERA-Interim in the Supplementary Material could be helpful.

5. I agree with RC1 that the role of Noah-Sat in the study is not explicitly stated. I understand it is used as a proxy for observations in Section 5, and that in Section 6 it helps assess the parameterizations in Noah and Noah-MP, since the surface properties should be ok. But I wonder if Section 2.1.1 could be improved to better explain these reasons why Noah-Sat is included.

6. I agree with RC1 that it is easiest to assess performance with maps of differences (e.g., Figs. 2–5, and Figs. 7–9).

7. Is this the first assessment of all these LSMs in Africa using WRF? If so, state it somewhere. If not, include references.

8. In the end, it was not clear why the regionalization of Africa occurred. In Section 4.3 I learn that it is for the Dinamica EGO predictions, however in Section 3.2 I was told it was for the LAI monthly profiles. Is it both? If so, connect the two ideas.

9. The study prefers to use Dinamica EGO output instead of MODIS directly to reduce "noisiness". Is this a usual practice? Is there some previous reference?

10. How different is the "default" land use in the validation and the MODIS 2001? Is "validation 2013" very different from "year 2013 of LU01"?

11. Is Fig. 10 similar to other results of LULCC studies in Africa?

12. I do not know how to interpret Table 4. Please explain what is "First Region" and "Second Region".

**Technical details**

1. As pointed out in RC1, some acronyms are not explained. I point out RL. Also in Table 9 it says WS10, but in text it is used as WSP10.

2. Abstract: "climate signals" is used too often. I wonder if "illogical" is the right word.

3. Text sections in supplementary material should be labelled, to be referred to from the main text easily (e.g., ST1, ST2).

4. Check the use of "en dash" for ranges of numbers (e.g., domain ranges).

5. I would stick to "DJF" for winter season, rather than "JFD".

6. Authors prefer the use of block titles for referring to panels in figures, instead of using labels (e.g., (a), (b)). It is ok, but I think the font size could be reduced. Also figures should use the same font types (titles are serif, but coordinates are sans-serif).

7. Tables are missing punctuation signs in captions and footnotes. I also think that tables with a lot of text are more readable if left-aligned (raggedright).

8. Sometimes too many references are used. For instance, the number of applications of Dinamica EGO has 12 references.

9. Figure captions in supplementary material say "verses" instead of "versus".

10. I wonder if the inset boxes in all maps with extreme and mean values are really being used. I like the idea, but they are not being used in the text or discussion.

11. Is Table S3 really necessary? Maybe only referencing Friedl, et al. 2002 is enough. By the way, this reference is nowhere in the references list.

**Line comments**

L38: through A land surface model

L39: different SIMULATED climate responses

L42: and IN the PRESCRIPTION OF

L47: with WIDESPREAD surface heterogeneity

L62: Air Force AND Hydrology Lab (Noah)

L63: reanalyses (maybe plural?)

L86: parametrization or parameterization?

L95: each year, EVERY

L96: model year is SIMULATED

L115: why "In"? Maybe only: "Noah and Noah-Sat are"

L127: also? why also?

L153: "(Sellers 1985)", check citation format

L173: why not "PFT distributions" instead of full name?

L174: "3 arc minute": a magnitude and a unit

L180: why not: categories 2, 4–10, 12 and 14

L183: "100%" instead of "one hundred percent"

L203: spacing: "Fig. 1", and "). These"

L204: parametrizations or parameterizations?

L210: "80%" instead of "eighty percent"

L218: "60%"

L219: "60%"

L222: "satellite-derived" (hyphen)

L276: why not LULC?

L277: experiment name is LULCC (double C)

L283: remove "simple or complex"

L289: "500 m" (why say "meter")

L292: I wonder if "ingested by" is the right word.

L327: "(Hagen 2003)", check citation format

L336: performance IS shown ("a list", singular)

L355: Figures 2–4 (because it starts a sentence)

L378: I would not reference Fig. 10 so far in advance.

L387: This leads TO an overall

L427: dataset INDICATES that (closer inspection, third person)

L442: "LSM, but provides" why but? maybe "and"

L443: wettest?

L448: "a more muted" maybe use "reduced" or synonym

L450: remove "select" and "very"

L459: wet instead of "wetter"

L478: Dinamica EGO

L524: patterns

Figs. 5, 7 and 9: I would add "Year 2013" to start the caption.

Fig. S11: has low text quality.

Table 9: It says Dowelling, instead of Downwelling

———————————————————

---

## Author Comment (AC1) · 25 Mar 2021

**Limitations of WRF land surface models for simulating land use and land cover change in Sub-Saharan Africa and development of an improved model (CLM-AF v. 1.0)**

Timothy Glotfelty, Diana Ramírez-Mejía, Jared Bowden, Adrian Ghilardi, and J. Jason West

**Replies to Referee 1**

Glotfelty et al. test the applicability of different LSMs incorporated in WRF RCM for Sub-Saharan Africa and their suitability to simulate the effects of land use changes on the regional climate conditions adequately. The results show that surface albedo, leaf area index and surface roughness are not accurately represented in the default models. Therefore, a new version of WRF coupled to the CLM LSM is developed, adjusted to the specific surface conditions in Sub-Saharan Africa. The topic of the study is within the scope of GMD and relevant for the large WRF modeling community in Africa and even beyond in the context of land use change effects on the regional climate in Africa. The manuscript is well structured and comprehensively written. The motivation of the paper is clear and the methods are well documented. Nevertheless, I have some concerns regarding the experimental setup, which need to be addressed by the authors before the manuscript is suitable for publication.

Response: We thank the reviewer for their comments on this work. Our point-by-point responses to the reviewer's comments can be found below.

**Major comments:**

1) In a first step, a model validation experiment is performed, in which simulations with five different models are conducted for one year. In general, I would say one year simulations are rather short to validate model performances, especially when only annual averages are presented. The authors argue that they chose the year 2013 because its a neutral year for the El Nino Southern Oscilation, but they do not really explain why this is the ideal boundary condition to validate the LSM performances. In any case, it is a single year that cannot represent the whole climate variability. Deviating atmospheric circulation conditions can considerably affect the impact of the land surface conditions on the regional climate conditions. Moreover, due to the exclusive consideration of annual averages, seasonal conditions are excluded in which the land surface conditions have larger impacts on the regional climate (e.g. dry conditions). Therefore, I recommend to extend the simulation period, or at least, to consider seasonal effects.

Response: We thank the reviewer for their comment. We understand the reviewer's concerns. The model validation experiment is meant to serve as an "out-of-the-box" meteorological comparison/evaluation of the WRF LSMs to illustrate the meteorological impacts of the default LSM deficiencies, rather than a full regional climate evaluation of the different model configurations. The rationale for this is that the WRF user community tends to use WRF in the "out-of-the-box" mode with minimal adjustments, for which the land cover/land use are

static and thus surface parameters are unchanging. To make this clearer, we have stated the proof-of-concept nature of our model evaluation more clearly in Section 4.1 and renamed the model validation experiment to the "meteorological evaluation experiment" (See below).

Lines 266-257 revised manuscript:

"The **meteorological evaluation** experiment consists of five simulations conducted for the year 2013, each using one of the five LSM configurations discussed above.  The year 2013 is selected because it is a neutral year for the El Niño Southern Oscillation **(ENSO) and thus should be representative of the mean state of Sub-Saharan Africa's ENSO climate variability**. While a single year comparison does not yield climate relevant statistics, it is sufficient to demonstrate differences in the meteorology between the five LSM configurations and the mechanisms responsible for these differences. **This is because the prescribed surface parameters from the LSM do not vary between years and thus the impact from these parameters on the simulated meteorology will be similar (or at least the impact from each LSM will remain similar relative to the others) regardless of the model's overall meteorological state.** The **meteorological evaluation** simulations are conducted with default greenhouse gas concentrations and MODIS 21 class land use data. These default settings are chosen to illustrate the performance that can be expected from the publicly available WRF model."

 Our overall goal with this experiment is to compare the impact of these surface parameters in different LSMs on the model's meteorological performance. Since the surface parameters do not vary between years within the WRF LSMs, the impact of the surface parameters from the LSMs will be similar relative to one another no matter which model year is simulated. For example, we find that the Noah LSM's albedo is strongly overestimated and this results in underpredicted surface temperatures. Therefore, no matter which meteorological years are simulated (i.e., whether that year is drier and less cloudy or moister and cloudier) the surface will always be too reflective and the surface energy balance will always be biased towards lower temperatures. While it is within the realm of possibility that in a particularly cold year the Noah LSM could perform better than the other LSMs, this would not be a meaningful result because the better agreement would be the result of an incorrect albedo. For these reasons, we do not think that extending the simulated period for the meteorological evaluation experiment would yield any more robust conclusions to justify the computational expense.

Per the reviewer's suggestion we have included seasonal surface plots in supplementary material (to save on space in the main text) to consider the seasonal evaluation of upwelling surface radiation at the Earth's surface (USRS), 2-m temperature (T2), and precipitation (See below). Additionally, references to these figures in the main text have been added (See bold and underlined below).

Lines 451-452 revised manuscript:

"Annual average spatial plots of USRS compared with CERES-EBAF estimates are shown in Fig. 7, **with seasonal average spatial plots shown in Fig. S5 of the supplementary material**."

Lines 468-459 revised manuscript:

"To understand the impact of surface parameters on near surface temperatures, the spatial plots of annual average T2 compared with CRU estimates are shown in Fig. 9, **with seasonal spatial plots shown in Fig. S10 of the supplementary material**."

Lines 507-508 revised manuscript:

"**Additionally, seasonal spatial plots of PRE compared with TRMM and annual average differences between TRMM and the LSMs and shown in Fig. S16 and Fig. S17, respectively**. All LSM simulations reasonably capture the annual (Fig. 11) **and seasonal (Fig. S16)** spatial patterns and magnitude of PRE."

[Figure]

Fig S5: 2013 seasonal average upwelling shortwave radiation at the Earth's surface (W m$^{-2}$) for CERES-EBAF estimates and WRF

[Figure]

Fig S10: 2013 seasonal average 2-m temperature (°C) for CRU estimates and WRF

[Figure]

Fig S16: 2013 seasonal average precipitation (mm day⁻¹) for TRMM estimates and WRF

Based on the rationale mentioned above, the reason we chose an ENSO neutral year is because neutral/transition ENSO conditions are the most common (e.g., https://origin.cpc.ncep.noaa.gov/products/analysis_monitoring/ensostuff/ONI_v5.php) and thus this phase of the climate variability is the most representative of the mean state of Sub-Sahran Africa's ENSO climate variability. We have added this to the rationale in the main text (Lines 267-268 revised manuscript):

"**The year 2013 is selected because it is a neutral year for the El Niño Southern Oscillation and thus should be representative of the mean state of Sub-Saharan Africa's ENSO climate variability**".

2) In a second step, it is intended to quantify the impact of land use changes on the simulation results with different LSMs. For this, the results of climate simulations for the period 2001-2010 with static land use conditions are compared to results of climate simulations for the period 2010-2015, including observed land use changes. But differences between two simulations with different land use conditions do not have to be caused inevitably/exclusively by the different land use conditions, in the case of deviating simulation periods. The different atmospheric circulation conditions in both periods can have certain impacts on the simulation results. Thus, from my point of view, identical simulation periods would have been preferable (2001-2015). If it is not possible to perform these simulations with respect to computing time, one could eventually reduce the number of LSMs based on the results of the validation experiment. By the way, I do not really understand why Noah Sat is included in the study, if one cannot use it for land use change scenarios at all. The authors should at least discuss the potential effects of the different simulation periods.

Response: We thank the reviewer for their comment. There appears to be some confusion with regards to the land use impact experiments. In both simulations (i.e., LU01 and LUD) we are simulating the same six-year meteorological period of 2010 to 2015, we do not simulate the years 2001-2009 in either experiment. Therefore, the atmospheric circulation in both periods are identical as forced by the initial and boundary conditions, with the differences resulting only from the land use and land cover.

These simulations represent the effect of land use and land cover change since the year 2001. The LU01 simulations uses static LULC representing the year 2001 from MODIS but meteorology representative of the years 2010-2015, and the LUD experiment uses both LULC from Dinamica EGO and meteorology for the years 2010-2015.

Since understanding our simulation setup is critical, we have made significant revisions to improve how this is communicated (see underlined below).

Lines 277-291 revised manuscript:

"The LULCC experiment simulates **recent climate responses from LULCC since the year 2001 by comparing simulations with static LULC from 2001 with dynamic LULC representing 2010-2015. In both cases, meteorology is simulated for the six-year period of 2010-2015. These** two simulations differing in LULCC **are conducted for each LSM configuration, using the Noah, Noah-MP, CLM-D, and CLM-AF LSMs**. The first simulation for each LSM uses static LULC from **MODIS representing** the year 2001 for each simulated year (i.e., 2010-2015), hereafter referred to as LU01. The second uses dynamic LULC **from the MODIS 21 class land use dataset that is processed by the Dinamica EGO land use modeling framework (Soares-Filho et al., 2002 – described in more detail below)** for each simulated year **in the 2010-2015 period**, hereafter referred to as LUD. The **six-year average** differences between the LU01 and LUD simulations delineate the climate response to LULCC. **The time period 2010-2015 is selected because it is far enough away from the year 2001 to show significant impacts from LULCC and because it contains the full ENSO climate variability cycle.** Noah-Sat is excluded because LAI and albedo parameters derived from satellite data could be impacted by climatological variability**, and therefore do not only represent** LULCC. The LULCC simulations also utilize global average greenhouse gas concentrations for each simulation year **(2010-2015)** from the National Oceanic and Atmospheric Administration's (NOAA) Earth System Research Laboratory (ESRL) Global Monitoring Division. In the LULCC experiment, each year is a discreet simulation with a 3-month spin-up in which the model LULC is updated at the start of each year. This is necessary because the WRF modelling framework treats LULC as a static field."

As stated, it is difficult to use Noah-Sat to do a LULCC experiment, which is why it is not included in this second experiment (e.g., Figures 13-16). However, Noah-Sat is important for the meteorological evaluation experiment because it is the LSM configuration in WRF with the most accurate LAI and albedo parameters, which allows it to serve as a pseudo observation to compare the LSM parameters against. We have added this discussion to Section 2.1.1 in accordance with the suggestions of reviewer 2 (Lines 130-132 revised manuscript):

"**However, Noah-Sat is useful for meteorological evaluations, because it has the most accurate surface parameters in the current WRF modelling system. Therefore, Noah-Sat can be used as pseudo-observations to understand deficiencies in the surface parameter methodologies of the other WRF LSMs.**"

3) It is very difficult to assess the differences between the different LSMs in the validation experiment based on the shown figures. It is therefore very difficult to compare these differences to the changes caused by land use changes. Plots of the differences to observations as shown in section 7 would help a lot.

Response: We thank the reviewer for their comment. We evaluated model performance in the meteorological evaluation experiment through both soccer goal plots and spatial plots. The soccer goal plots provide insights into the model performance at both a domain-wide and regional level that convey to the reader how far off from the observations the different

parameters are. The spatial plots display how well the different model configurations capture the spatial patterns and magnitudes of our three key variables of interest (i.e., USRS, T2, and precipitation). Because we think these elements of the evaluation are the most relevant we have elected to keep these as the model evaluation figures in the main text. However, to assist readers and the reviewer we have also included difference plots of our three key variables in the supplementary material with citations in the main text (See below).

Lines 452-253 revised manuscript:

"**Additionally, annual average difference plots with CERES-EBAF for each LSM are shown in Fig. S6.**"

Lines 469-470 revised manuscript:

"**Annual average differences between CRU and the LSMs are also shown in Fig. S11**."

Lines 472-473 revised manuscript:

"The only clear impact of surface albedo inaccuracy on annual average T2 is the relatively stronger cold bias in the Noah LSM (Fig. 9, **Fig. S11**)."

Lines 507-508 revised manuscript:

"**Additionally, seasonal spatial plots of PRE compared with TRMM and annual average differences between TRMM and the LSMs and shown in Fig. S16 and Fig. S17, respectively.**"

Lines 510-512 revised manuscript:

"The greatest underpredictions occur in arid regions (ND, ED, SD, NESD, and WSD) and portions of East Africa (EM, CM, and LVW), while regions in South Africa (SSD and SM) and EW typically experience the strongest overprediction across the LSMs (Fig. 10, Fig. S12, **Fig. S17**)."

[Figure]

Fig S6: 2013 annual average differences in upwelling shortwave radiation at the Earth's surface (W m⁻²) between the WRF simulations and CERES-EBAF estimates (WRF – CERES-EBAF)

[Figure]

Fig S11: 2013 annual average differences in 2-m temperature (°C) between the WRF simulations and CRU estimates (WRF-CRU)

[Figure]

Fig S17: 2013 annual average differences in precipitation (mm day$^{-1}$) between the WRF simulations and TRMM estimates (WRF – TRMM)

4) To be able to understand the results of the validation experiment comprehensively, an assessment of the sensible and latent heat fluxes is necessary.

Response: We thank the reviewer for their comment. Per the reviewer's suggestion, we have included new Figures 5 and 6 that contain spatial plot comparisons of the annual average latent heat and sensible fluxes. Additionally, we have added the following discussion to the end of the results section 5 to discuss the differences in these parameters.

Lines 436-448 revised manuscript:

"For both latent (LH) and sensible (HFX) fluxes (Fig. 5 and Fig. 6), all LSMs produce similar annual average spatial distributions.  LH are more similar amongst LSMs (Fig. 5), with the key difference being larger LH (~10–20 W m$^{-2}$) in the most heavily vegetated portions of the domain for the CLM-D and CLM-AF configurations.  The similar LH for CLM-D and CLM-AF suggests a mechanistic difference that may be related to the vegetation canopy approximation in CLM that does not account for gaps within the canopy or between vegetation crowns. However, the values are the largest for CLM-AF in regions containing savanna, likely due to the larger values of LAI in these regions during the drier seasons (Fig. 3).

For HFX (Fig. 6), the Noat-Sat LSM produces the largest fluxes, especially in the semi-dry regions of eastern and southern Africa. This is likely a combination of Noah-Sat having the lowest albedo in vegetated regions leading to more surface energy absorption and Noah-Sat having consistently low LAI values in these regions throughout the year compared to other LSMs (Fig. 2 and Fig. 3). Both CLM-D and CLM-AF have lower HFX compared to the other

LSMs in vegetated areas, again likely due to the vegetation canopy assumptions. However, CLM-D has higher HFX in southern Africa comparable to those of Noah and Noah-MP. This is likely the result of Noah, Noah-MP, and CLM-D having much larger than realistic fluctuations in LAI between the wetter and drier seasons in this region (Fig. 3)"

[Figure]

Figure 5: Comparison of annual average latent heat flux (W m$^{-2}$) between LSM configurations.

[Figure]

Figure 6: Comparison of annual average sensible heat flux (W m$^{-2}$) between LSM configurations.

**Minor comments:**

1) The biogeophysical effects of the surface roughness on the climate impacts of land use changes is not considered. For instance, this impact can be seen for the deforestation regions. The model results consistently show a warming with deforestation. In the manuscript, this is explained by reduced evapotranspiration rates and an associated reduced latent cooling. But if the reduced latent heat fluxes are the reason for the increased near-surface temperatures, accordingly the sensible heat fluxes should be increased (due to the increased temperature gradient between the land surface and the atmosphere). But this is not the case, the sensible heat fluxes are also reduced. Therefore, I suppose that the efficiency of the deforested land surface to transform the incoming solar energy in turbulent heat is reduced due to the reduced surface roughness, resulting in a warming of the surface (e.g. Winckler et al., 2019; Breil et al., 2020).

Response: We thank the reviewer for their comment, which provides a more complete explanation of the physical processes in the model. We have added several tables (S16-S27) in the supplementary material to include the full changes seen in the surface energy balance and near-surface temperature profiles. Additionally, we have made significant changes to Section 7.2 to account for the biogeophysical impacts of surface roughness length as shown below, and have revised the conclusions accordingly. However, even with the addition of this information it appears that the dominant factor controlling most of the warming and cooling predicted by the model is still increases or decreases in evaporative cooling.

Lines 579-658 revised manuscript:

[revised manuscript text omitted]

[*]: Shows average difference for a broad class of LULCC followed by the average difference in the major MODIS LULC transitions that comprise that class. MODIS Land Use Categories: 2 – Evergreen Broad Leaf Forest; 7 – Open Shrublands; 8 – Woody Savanna; 9 – Savannas; 10 – Grasslands; 12 – Croplands; 14 –Cropland/Natural Mosaic; 16 – Barren/ Sparsely Vegetated.

Table S17: Annual Average Surface Radiative Flux Change (W m$^{-2}$) in WRF Grid Cells that experience LULCCs between 2001 and 2015 with Noah-MP

| Transition | USRS Day | USRS Night | SWDOWN Day | SWDOWN Night | ULRS Day | ULRS Night | GLW Day | GLW Night |
|---|---|---|---|---|---|---|---|---|
| **Agricultural Expansion*** | 22.8 | - | 8.0 | - | 4.7 | 1.1 | -0.9 | 0.0 |
| 10 to 12 | -0.5 | - | 0.2 | - | 2.4 | 0.7 | -0.1 | 0.1 |
| 2 to 14 | 25.2 | - | 10.6 | - | 13.5 | 6.0 | -0.4 | 0.4 |
| 8 to 14 | 33.3 | - | 12.4 | - | 4.9 | 0.8 | -1.5 | -0.1 |
| 10 to 14 | -0.7 | - | 0.4 | - | 0.8 | 0.4 | -0.1 | 0.0 |
| **Deforestation/Degradation*** | 9.1 | - | 1.9 | - | 1.6 | -0.1 | -0.3 | -0.1 |
| 8 to 9 | 9.7 | - | 2.3 | - | -0.1 | -0.1 | -0.3 | -0.1 |
| 9 to 7 | -3.1 | - | -2.1 | - | 5.6 | -0.9 | -0.1 | -0.3 |
| 9 to 10 | 20.7 | - | 4.4 | - | 2.3 | -0.3 | -0.6 | -0.1 |
| **Greening*** | -6.3 | - | -1.1 | - | -3.7 | -0.9 | 0.0 | -0.1 |
| 9 to 8 | -8.9 | - | -1.9 | - | -0.1 | -0.1 | 0.1 | -0.2 |
| 10 to 9 | -25.2 | - | -7.5 | - | -3.4 | -0.5 | 0.7 | -0.4 |
| 16 to 7 | -5.5 | - | -1.2 | - | -4.0 | -0.6 | 0.1 | 0.2 |
| 16 to 10 | 1.6 | - | 0.6 | - | -5.1 | -1.9 | -0.2 | 0.1 |

*: Shows average difference for a broad class of LULCC followed by the average difference in the major MODIS LULC transitions that comprise that class. MODIS Land Use Categories: 2 – Evergreen Broad Leaf Forest; 7 – Open Shrublands; 8 – Woody Savanna; 9 – Savannas; 10 – Grasslands; 12 – Croplands; 14 –Cropland/Natural Mosaic; 16 – Barren/ Sparsely Vegetated.

Table S18: Annual Average Surface Radiative Flux Change (W m$^{-2}$) in WRF Grid Cells that experience LULCCs between 2001 and 2015 with CLM-D

| Transition | USRS Day | USRS Night | SWDOWN Day | SWDOWN Night | ULRS Day | ULRS Night | GLW Day | GLW Night |
|---|---|---|---|---|---|---|---|---|
| **Agricultural Expansion*** | 17.4 | - | 6.2 | - | -10.5 | 3.8 | -0.6 | 0.3 |
| 10 to 12 | 3.3 | - | 1.4 | - | -11.6 | 8.2 | 0.1 | 0.4 |
| 2 to 14 | 12.9 | - | 5.0 | - | 11.4 | 11.4 | 0.0 | 1.0 |
| 8 to 14 | 29.8 | - | 9.4 | - | -15.8 | 1.3 | -0.9 | 0.2 |
| 10 to 14 | 1.4 | - | 0.8 | - | -6.0 | 4.1 | -0.2 | 0.1 |
| **Deforestation/Degradation*** | 11.1 | - | 1.4 | - | -2.6 | -0.3 | -0.1 | -0.1 |
| 8 to 9 | 18.9 | - | 2.1 | - | -6.4 | -2.6 | 0.0 | -0.3 |
| 9 to 7 | -7.3 | - | -3.1 | - | 2.2 | 3.0 | -0.1 | 0.3 |
| 9 to 10 | 8.6 | - | 2.8 | - | -2.0 | 2.1 | -0.4 | 0.2 |
| **Greening*** | -19.2 | - | -3.3 | - | 7.9 | -6.3 | 0.2 | -0.4 |
| 9 to 8 | -18.7 | - | -2.6 | - | 4.9 | 1.4 | -0.1 | 0.1 |
| 10 to 9 | -2.8 | - | -6.8 | - | 0.9 | -4.3 | 1.0 | -0.7 |
| 16 to 7 | -37.5 | - | -7.3 | - | 26.9 | -11.6 | 1.0 | -0.8 |
| 16 to 10 | -21.2 | - | -3.8 | - | 19.4 | -20.8 | 0.1 | -1.4 |

*: Shows average difference for a broad class of LULCC followed by the average difference in the major MODIS LULC transitions that comprise that class.  MODIS Land Use Categories: 2 – Evergreen Broad Leaf Forest; 7 – Open Shrublands; 8 – Woody Savanna; 9 – Savannas; 10 – Grasslands; 12 – Croplands; 14 –Cropland/Natural Mosaic; 16 – Barren/ Sparsely Vegetated.

Table S19: Annual Average Surface Radiative Flux Change (W m$^{-2}$) in WRF Grid Cells that experience LULCCs between 2001 and 2015 with Noah

| Transition | USRS Day | USRS Night | SWDOWN Day | SWDOWN Night | ULRS Day | ULRS Night | GLW Day | GLW Night |
|---|---|---|---|---|---|---|---|---|
| **Agricultural Expansion*** | -10.3 | - | -2.2 | - | 0.5 | -1.2 | 0.3 | 0.1 |
| 10 to 12 | -3.5 | - | -1.1 | - | 1.4 | -0.2 | 0.1 | 0.1 |
| 2 to 14 | 37.3 | - | 11.9 | - | -1.6 | -2.0 | -0.5 | -0.1 |
| 8 to 14 | -32.2 | - | -7.6 | - | -1.1 | -1.9 | 0.7 | 0.3 |
| 10 to 14 | -2.1 | - | -1.1 | - | 0.8 | -0.4 | 0.2 | 0.1 |
| **Deforestation/Degradation*** | -12.4 | - | -2.3 | - | -0.4 | -0.5 | 0.3 | 0.1 |
| 8 to 9 | -35.0 | - | -6.7 | - | -1.6 | -0.9 | 0.8 | 0.1 |
| 9 to 7 | 29.0 | - | 5.2 | - | 2.5 | 1.1 | -0.7 | 0.1 |
| 9 to 10 | 4.0 | - | 0.2 | - | 1.1 | -0.2 | 0.0 | 0.1 |
| **Greening*** | -18.0 | - | -2.6 | - | 3.7 | 0.4 | 0.3 | -0.1 |
| 9 to 8 | 34.8 | - | 5.6 | - | 2.5 | 0.9 | -0.4 | 0.2 |
| 10 to 9 | -5.4 | - | -1.7 | - | -1.2 | 0.2 | 0.3 | 0.2 |
| 16 to 7 | -59.2 | - | -8.0 | - | 7.6 | -0.2 | 0.9 | -0.5 |
| 16 to 10 | -82.8 | - | -12.7 | - | 5.7 | -0.7 | 1.4 | -0.7 |

*: Shows average difference for a broad class of LULCC followed by the average difference in the major MODIS LULC transitions that comprise that class. MODIS Land Use Categories: 2 – Evergreen Broad Leaf Forest; 7 – Open Shrublands; 8 – Woody Savanna; 9 – Savannas; 10 – Grasslands; 12 – Croplands; 14 –Cropland/Natural Mosaic; 16 – Barren/ Sparsely Vegetated.

Table S20: Annual Average Surface Heat Flux Change (W m$^{-2}$) in WRF Grid Cells that experience LULCCs between 2001 and 2015 with CLM-AF

| Transition | HFX | | LH | | GRDFLX | |
|---|---|---|---|---|---|---|
| | Day | Night | Day | Night | Day | Night |
| **Agricultural Expansion*** | 0.6 | 1.2 | -9.0 | 1.1 | 4.0 | -3.8 |
| 10 to 12 | 0.9 | 0.4 | -6.0 | 0.4 | 2.1 | -2.1 |
| 2 to 14 | 0.4 | 9.7 | -61.0 | 8.7 | 31.1 | -30.1 |
| 8 to 14 | -1.4 | 1.6 | -8.7 | 0.9 | 3.8 | -3.7 |
| 10 to 14 | 10.0 | -1.0 | -4.3 | -0.4 | -1.3 | 1.1 |
| **Deforestation/Degradation*** | -6.7 | 2.2 | -8.3 | 0.7 | 5.3 | -5.1 |
| 8 to 9 | -4.7 | 1.6 | -6.4 | 0.4 | 3.7 | -3.6 |
| 9 to 7 | -14.7 | 2.6 | -10.1 | 0.8 | 7.5 | -7.1 |
| 9 to 10 | -9.3 | 2.4 | -4.2 | 0.4 | 4.6 | -4.3 |
| **Greening*** | 8.4 | -3.8 | 13.3 | -1.4 | -9.0 | 8.7 |
| 9 to 8 | 3.0 | -1.8 | 7.4 | -0.6 | -3.4 | 3.2 |
| 10 to 9 | 13.8 | -4.3 | -0.1 | -1.3 | -9.8 | 8.8 |
| 16 to 7 | 10.0 | -2.8 | 4.7 | -0.3 | -6.1 | 5.8 |
| 16 to 10 | 27.3 | -3.9 | 10.3 | -0.5 | -11.9 | 11.8 |

*: Shows average difference for a broad class of LULCC followed by the average difference in the major MODIS LULC transitions that comprise that class. MODIS Land Use Categories: 2 – Evergreen Broad Leaf Forest; 7 – Open Shrublands; 8 – Woody Savanna; 9 – Savannas; 10 – Grasslands; 12 – Croplands; 14 –Cropland/Natural Mosaic; 16 – Barren/ Sparsely Vegetated.

Table S21: Annual Average Surface Heat Flux Change (W m$^{-2}$) in WRF Grid Cells that experience LULCCs between 2001 and 2015 with Noah-MP

| Transition | HFX | | LH | | GRDFLX | |
|---|---|---|---|---|---|---|
| | Day | Night | Day | Night | Day | Night |
| **Agricultural Expansion*** | -20.1 | 1.9 | -3.3 | 0.0 | 3.0 | -2.9 |
| 10 to 12 | -3.6 | 0.6 | 1.0 | -0.4 | 0.9 | -0.8 |
| 2 to 14 | -4.2 | 6.6 | -38.6 | 1.9 | 14.0 | -13.8 |
| 8 to 14 | -30.8 | 2.2 | 0.6 | -0.1 | 3.0 | -3.0 |
| 10 to 14 | -2.1 | 0.2 | 1.8 | -0.2 | 0.4 | -0.4 |
| **Deforestation/Degradation*** | -7.6 | 0.9 | -2.3 | -0.1 | 0.8 | -0.8 |
| 8 to 9 | -7.6 | 0.3 | -0.2 | 0.0 | 0.3 | -0.3 |
| 9 to 7 | -2.2 | 1.9 | -3.6 | -0.1 | 1.1 | -1.1 |
| 9 to 10 | -19.1 | -1.0 | -0.2 | 1.0 | 1.1 | -1.1 |
| **Greening*** | 1.5 | -1.3 | 9.4 | 0.0 | -2.0 | 2.0 |
| 9 to 8 | 6.5 | -0.4 | 0.7 | 0.1 | -0.2 | 0.2 |
| 10 to 9 | 24.6 | -1.8 | -0.7 | 0.0 | -2.2 | 1.9 |
| 16 to 7 | 2.2 | -0.7 | 7.8 | 0.1 | -1.4 | 1.4 |
| 16 to 10 | -2.9 | -1.3 | 10.2 | 0.3 | -3.1 | 2.9 |

*: Shows average difference for a broad class of LULCC followed by the average difference in the major MODIS LULC transitions that comprise that class.  MODIS Land Use Categories: 2 – Evergreen Broad Leaf Forest; 7 – Open Shrublands; 8 – Woody Savanna; 9 – Savannas; 10 – Grasslands; 12 – Croplands; 14 –Cropland/Natural Mosaic; 16 – Barren/ Sparsely Vegetated.

Table S22: Annual Average Surface Heat Flux Change (W m$^{-2}$) in WRF Grid Cells that experience LULCCs between 2001 and 2015 with CLM-D

| Transition | HFX Day | HFX Night | LH Day | LH Night | GRDFLX Day | GRDFLX Night |
|---|---|---|---|---|---|---|
| **Agricultural Expansion***  | -8.8 | 1.9 | -8.1 | 0.2 | 3.9 | -3.8 |
| 10 to 12 | 0.0 | 0.4 | -3.1 | -1.2 | 0.3 | -0.3 |
| 2 to 14 | -3.2 | 12.3 | -63.0 | 6.2 | 31.9 | -30.7 |
| 8 to 14 | -12.1 | 0.2 | -2.8 | -0.5 | -0.5 | 0.5 |
| 10 to 14 | 0.8 | 0.1 | -0.5 | -0.8 | -0.3 | 0.3 |
| **Deforestation/Degradation***  | -4.5 | 0.3 | -2.9 | 0.0 | 0.3 | -0.4 |
| 8 to 9 | -1.9 | -2.0 | -0.1 | -0.6 | -4.7 | 4.4 |
| 9 to 7 | -6.0 | 1.9 | 0.5 | 0.2 | 3.3 | -3.3 |
| 9 to 10 | -14.0 | 3.2 | -2.3 | 0.2 | 6.0 | -5.7 |
| **Greening***  | 7.8 | -3.2 | 16.6 | -0.7 | -7.6 | 7.4 |
| 9 to 8 | -0.7 | 1.6 | 5.1 | 0.8 | 3.8 | -3.5 |
| 10 to 9 | 16.8 | -5.0 | -1.1 | -1.0 | -11.5 | 10.4 |
| 16 to 7 | 24.7 | -4.9 | 13.6 | 0.1 | -10.1 | 9.8 |
| 16 to 10 | 13.3 | -4.0 | 17.5 | -0.8 | -13.7 | 13.5 |

*: Shows average difference for a broad class of LULCC followed by the average difference in the major MODIS LULC transitions that comprise that class.  MODIS Land Use Categories: 2 – Evergreen Broad Leaf Forest; 7 – Open Shrublands; 8 – Woody Savanna; 9 – Savannas; 10 – Grasslands; 12 – Croplands; 14 –Cropland/Natural Mosaic; 16 – Barren/ Sparsely Vegetated.

Table S23: Annual Average Surface Heat Flux Change (W m$^{-2}$) in WRF Grid Cells that experience LULCCs between 2001 and 2015 with Noah

| Transition | HFX | | LH | | GRDFLX | |
|---|---|---|---|---|---|---|
| | Day | Night | Day | Night | Day | Night |
| **Agricultural Expansion**[*] | 5.4 | -0.7 | 5.5 | -1.0 | 3.1 | -3.0 |
| 10 to 12 | -0.8 | 0.4 | 2.3 | -0.2 | 0.2 | -0.2 |
| 2 to 14 | -0.4 | 1.2 | -21.2 | -2.0 | 2.8 | -2.7 |
| 8 to 14 | 11.9 | -2.0 | 20.1 | -1.4 | 5.8 | -5.7 |
| 10 to 14 | -2.3 | 0.5 | 3.0 | -0.2 | 0.3 | -0.3 |
| **Deforestation/Degradation**[*] | 9.3 | -1.1 | 3.7 | -0.5 | 2.2 | -2.2 |
| 8 to 9 | 23.9 | -2.9 | 11.4 | -0.8 | 4.7 | -4.6 |
| 9 to 7 | -23.2 | 2.8 | -7.8 | 0.2 | -4.0 | 3.9 |
| 9 to 10 | -3.8 | -0.1 | -0.2 | -0.4 | 0.8 | -0.8 |
| **Greening**[*] | 12.2 | -1.0 | 0.0 | 0.2 | 0.2 | -0.3 |
| 9 to 8 | -16.6 | 2.8 | -19.7 | 0.7 | -4.3 | 4.2 |
| 10 to 9 | 6.2 | 0.1 | -1.7 | 0.4 | -0.7 | 0.6 |
| 16 to 7 | 42.5 | -3.4 | 5.9 | -0.6 | 3.8 | -3.9 |
| 16 to 10 | 57.6 | -7.8 | 16.6 | -0.4 | 8.2 | -8.3 |

*: Shows average difference for a broad class of LULCC followed by the average difference in the major MODIS LULC transitions that comprise that class. MODIS Land Use Categories: 2 – Evergreen Broad Leaf Forest; 7 – Open Shrublands; 8 – Woody Savanna; 9 – Savannas; 10 – Grasslands; 12 – Croplands; 14 –Cropland/Natural Mosaic; 16 – Barren/ Sparsely Vegetated.

Table S24: Annual Average Near Surface Temperature Profile Change in WRF Grid Cells that experience LULCCs between 2001 and 2015 with CLM-AF

| Transition | TSK ($°C$) | | T2 ($°C$) | | TATM ($10^{-1}$ $°C$) | | TGSATM ($10^{-2}$ $°C$ $m^{-1}$) | |
|---|---|---|---|---|---|---|---|---|
| | Day | Night | Day | Night | Day | Night | Day | Night |
| **Agricultural Expansion*** | 0.7 | 0.1 | 0.0 | 0.3 | 0.1 | 0.7 | 2.2 | 0.0 |
| 10 to 12 | 0.7 | 0.0 | 0.1 | 0.2 | 0.2 | 0.2 | 2.1 | -0.3 |
| 2 to 14 | 3.1 | 1.8 | 0.4 | 2.3 | 1.8 | 6.3 | 9.6 | 2.0 |
| 8 to 14 | 0.5 | 0.2 | 0.0 | 0.3 | -0.1 | 0.5 | 1.8 | 0.3 |
| 10 to 14 | 0.8 | -0.3 | 0.1 | 0.0 | 0.7 | -0.2 | 2.5 | -0.9 |
| **Deforestation/Degradation*** | 0.0 | 0.5 | 0.0 | 0.4 | -0.2 | 1.4 | 0.2 | 0.8 |
| 8 to 9 | 0.5 | 0.2 | 0.0 | 0.3 | -0.2 | 0.8 | 1.9 | 0.1 |
| 9 to 7 | -1.9 | 1.9 | -0.1 | 0.8 | -0.2 | 4.0 | -6.4 | 4.0 |
| 9 to 10 | -0.4 | 0.3 | -0.1 | 0.3 | -0.7 | 0.7 | -1.1 | 0.5 |
| **Greening*** | 1.1 | -1.4 | -0.1 | -0.7 | -1.3 | -2.5 | 4.6 | -3.1 |
| 9 to 8 | -0.4 | -0.2 | 0.0 | -0.2 | 0.2 | -0.4 | -1.5 | -0.2 |
| 10 to 9 | 0.5 | -0.6 | 0.1 | -0.6 | 0.7 | -2.5 | 1.4 | -0.3 |
| 16 to 7 | 4.0 | -2.0 | -0.3 | -0.5 | -3.7 | -2.9 | 16.2 | -5.0 |
| 16 to 10 | 4.8 | -4.8 | -0.1 | -1.4 | -2.8 | -6.1 | 17.8 | -12.4 |

*: Shows average difference for a broad class of LULCC followed by the average difference in the major MODIS LULC transitions that comprise that class. MODIS Land Use Categories: 2 – Evergreen Broad Leaf Forest; 7 – Open Shrublands; 8 – Woody Savanna; 9 – Savannas; 10 – Grasslands; 12 – Croplands; 14 –Cropland/Natural Mosaic; 16 – Barren/ Sparsely Vegetated.

Table S25: Annual Average Near Surface Temperature Profile Change in WRF Grid Cells that experience LULCCs between 2001 and 2015 with Noah-MP

| Transition | TSK (°C) | | T2 (°C) | | TATM ($10^{-1}$ °C) | | TGSATM ($10^{-2}$ °C m$^{-1}$) | |
|---|---|---|---|---|---|---|---|---|
| | Day | Night | Day | Night | Day | Night | Day | Night |
| **Agricultural Expansion***  | 0.7 | 0.2 | 0.0 | 0.2 | -1.3 | 0.9 | 3.3 | 0.1 |
| 10 to 12 | 0.4 | 0.1 | 0.2 | 0.1 | 0.0 | 0.7 | 1.4 | 0.1 |
| 2 to 14 | 2.2 | 1.0 | 0.3 | 0.9 | 0.5 | 4.1 | 7.4 | 0.8 |
| 8 to 14 | 0.8 | 0.1 | -0.2 | 0.2 | -2.2 | 0.5 | 3.9 | 0.1 |
| 10 to 14 | 0.1 | 0.1 | 0.1 | 0.1 | -0.1 | 0.2 | 0.5 | 0.1 |
| **Deforestation/Degradation***  | 0.3 | 0.0 | -0.1 | 0.0 | -0.6 | -0.1 | 1.3 | 0.0 |
| 8 to 9 | 0.0 | 0.0 | -0.1 | 0.0 | -0.6 | -0.1 | 0.3 | 0.0 |
| 9 to 7 | 0.9 | -0.1 | -0.2 | 0.1 | -0.5 | -0.5 | 3.3 | -0.2 |
| 9 to 10 | 0.3 | -0.1 | -0.1 | 0.0 | -1.2 | -0.1 | 1.9 | -0.1 |
| **Greening***  | -0.6 | -0.2 | -0.1 | -0.2 | -0.1 | -0.1 | -2.0 | -0.5 |
| 9 to 8 | 0.0 | 0.0 | 0.0 | 0.0 | 0.3 | -0.1 | -0.2 | 0.1 |
| 10 to 9 | -0.5 | -0.1 | 0.2 | -0.2 | 0.1 | -0.1 | -2.6 | 0.2 |
| 16 to 7 | -0.6 | -0.1 | -0.1 | -0.1 | -0.1 | 0.9 | -2.2 | -1.1 |
| 16 to 10 | -0.8 | -0.3 | -0.1 | -0.3 | -0.4 | 0.8 | -2.5 | -1.7 |

*: Shows average difference for a broad class of LULCC followed by the average difference in the major MODIS LULC transitions that comprise that class. MODIS Land Use Categories: 2 – Evergreen Broad Leaf Forest; 7 – Open Shrublands; 8 – Woody Savanna; 9 – Savannas; 10 – Grasslands; 12 – Croplands; 14 –Cropland/Natural Mosaic; 16 – Barren/ Sparsely Vegetated.

Table S26: Annual Average Near Surface Temperature Profile Change in WRF Grid Cells that experience LULCCs between 2001 and 2015 with CLM-D

| Transition | TSK (°C) Day | TSK (°C) Night | T2 (°C) Day | T2 (°C) Night | TATM ($10^{-1}$ °C) Day | TATM ($10^{-1}$ °C) Night | TGSATM ($10^{-2}$ °C m$^{-1}$) Day | TGSATM ($10^{-2}$ °C m$^{-1}$) Night |
|---|---|---|---|---|---|---|---|---|
| **Agricultural Expansion*** | -2.0 | 0.8 | -0.1 | 0.3 | -0.5 | 1.5 | -6.6 | 1.7 |
| 10 to 12 | -2.4 | 1.8 | 0.1 | 0.2 | 1.6 | 2.4 | -9.3 | 4.7 |
| 2 to 14 | 1.3 | 1.9 | 0.4 | 2.3 | 1.2 | 6.4 | 3.9 | 2.3 |
| 8 to 14 | -2.7 | 0.3 | -0.2 | -0.1 | -1.3 | 0.3 | -8.3 | 0.8 |
| 10 to 14 | -1.3 | 0.9 | 0.0 | 0.1 | -0.1 | 0.8 | -4.3 | 2.7 |
| **Deforestation/Degradation*** | -0.4 | -0.1 | 0.0 | 0.0 | -0.3 | -0.2 | -1.0 | -0.1 |
| 8 to 9 | -0.7 | -0.5 | 0.0 | -0.4 | -0.2 | -1.3 | -2.3 | -0.7 |
| 9 to 7 | 0.0 | 0.6 | -0.1 | 0.4 | -1.3 | 0.9 | 0.8 | 1.5 |
| 9 to 10 | -0.6 | 0.3 | -0.1 | 0.3 | -1.1 | 0.9 | -1.5 | 0.6 |
| **Greening*** | 1.6 | -1.3 | 0.0 | -0.6 | -0.2 | -2.6 | 5.4 | -2.7 |
| 9 to 8 | 0.5 | 0.3 | 0.0 | 0.2 | -0.5 | 0.1 | 2.1 | 0.7 |
| 10 to 9 | 0.7 | -0.7 | 0.1 | -0.7 | 0.9 | -2.9 | 1.7 | -0.5 |
| 16 to 7 | 5.3 | -2.5 | 0.2 | -0.9 | 0.8 | -0.4 | 17.4 | -6.1 |
| 16 to 10 | 3.6 | -4.2 | 0.0 | -1.6 | -0.4 | -5.6 | 12.3 | -10.8 |

*: Shows average difference for a broad class of LULCC followed by the average difference in the major MODIS LULC transitions that comprise that class. MODIS Land Use Categories: 2 – Evergreen Broad Leaf Forest; 7 – Open Shrublands; 8 – Woody Savanna; 9 – Savannas; 10 – Grasslands; 12 – Croplands; 14 –Cropland/Natural Mosaic; 16 – Barren/ Sparsely Vegetated.

Table S27: Annual Average Near Surface Temperature Profile Change in WRF Grid Cells that experience LULCCs between 2001 and 2015 with Noah

| Transition | TSK (°C) Day | TSK (°C) Night | T2 (°C) Day | T2 (°C) Night | TATM ($10^{-1}$ °C) Day | TATM ($10^{-1}$ °C) Night | TGSATM ($10^{-2}$ °C m$^{-1}$) Day | TGSATM ($10^{-2}$ °C m$^{-1}$) Night |
|---|---|---|---|---|---|---|---|---|
| **Agricultural Expansion**[*] | -0.2 | -0.4 | 0.1 | -0.3 | 0.1 | -1.1 | -0.7 | -0.5 |
| 10 to 12 | 0.1 | -0.1 | -0.1 | -0.1 | -0.1 | -0.1 | 0.3 | -0.4 |
| 2 to 14 | -0.3 | -0.4 | -0.3 | -0.3 | -0.2 | -0.7 | -0.7 | -0.8 |
| 8 to 14 | -0.5 | -0.5 | 0.3 | -0.5 | 0.3 | -2.0 | -1.9 | -0.4 |
| 10 to 14 | 0.0 | -0.2 | -0.1 | -0.1 | -0.2 | -0.1 | 0.1 | -0.5 |
| **Deforestation/Degradation**[*] | -0.1 | -0.1 | 0.2 | -0.1 | 0.5 | -0.3 | -0.6 | -0.1 |
| 8 to 9 | -0.1 | -0.1 | 0.5 | -0.1 | 1.2 | -0.8 | -1.2 | 0.3 |
| 9 to 7 | 0.1 | 0.0 | -0.5 | 0.1 | -0.8 | 1.2 | 0.9 | -0.7 |
| 9 to 10 | -0.1 | -0.2 | -0.1 | -0.1 | -0.3 | -0.3 | -0.2 | -0.4 |
| **Greening**[*] | 0.3 | -0.1 | 0.2 | -0.1 | 0.9 | -0.5 | 0.4 | -0.1 |
| 9 to 8 | 0.3 | 0.1 | -0.3 | 0.1 | -0.6 | 0.6 | 1.4 | -0.2 |
| 10 to 9 | 0.2 | 0.3 | 0.1 | 0.2 | 0.4 | 0.9 | 0.4 | 0.3 |
| 16 to 7 | 0.5 | -0.5 | 0.5 | -0.5 | 2.6 | -1.7 | 0.0 | -0.7 |
| 16 to 10 | 0.2 | -0.5 | 0.8 | -0.6 | 3.1 | -3.1 | -1.3 | 0.2 |

[*]: Shows average difference for a broad class of LULCC followed by the average difference in the major MODIS LULC transitions that comprise that class. MODIS Land Use Categories: 2 – Evergreen Broad Leaf Forest; 7 – Open Shrublands; 8 – Woody Savanna; 9 – Savannas; 10 – Grasslands; 12 – Croplands; 14 –Cropland/Natural Mosaic; 16 – Barren/ Sparsely Vegetated.

2) simulation results show that the cloud cover is consistently overestimated in the validation experiment. At the same time, downward short-wave radiation (swdown?) is also overestimated. How does that fit together?

Response: We thank the reviewer for their comment. The cloud fraction (i.e. cloud cover) parameter only refers to the spatial extent of clouds, while the swdown parameter is determined by the cloud's optical thickness. For example, a model grid column that contains a single high-level cirrus cloud would have a cloud fraction of 100%, but that cloud would be very thin and not drastically reduce the swdown parameter. In these simulations, it is likely that the model is producing excess anvil clouds from the convection, resulting in the overpredicted cloud fraction. However, the clouds that do exist in the model are not sufficiently optically thick enough to reduce the swdown to the appropriate level. This can be seen in Figure S6 of the updated supplementary material, where the shortwave and longwave cloud forcing are underpredicted in all model configurations.

We have added the following sentence to the discussion of radiation variables for clarity:

Lines 465-467 revised manuscript: "**The underestimated cloud radiative forcing seems to indicate the model is not generating clouds of sufficient optical thickness, since cloud fractions are overestimated compared to satellite estimates (Fig. 10, Fig. S15).**"

3) Several abbreviations are used which are not explained in the text (e.g. SWDOWN, GLW, OLR, SWUPT).

Response: We thank the reviewer for their comment and bringing this to our attention. We found some inconsistencies in the names between the tables, supplementary material, and the main text. We have updated Table 9 below for consistency and added the full names in the main text before any abbreviations used.

Table 9: Evaluated Variables and Evaluation Datasets

| Variable | Acronym | Evaluation Dataset |
|---|---|---|
| 2-m Temperature | T2 | CRU TS4.02 and NCDC-ISD |
| Daily Maximum Temperature | T2MAX | CRU TS4.02 |
| Daily Minimum Temperature | T2MIN | CRU TS4.02 |
| Diurnal Temperature Range | DTR | CRU TS4.02 |
| 2-m Vapor Pressure | E2 | CRU TS4.02 |
| 2-m Dew point Temperature | $T_d2$ | NCDC-ISD |
| Precipitable Water Vapor | PWV | MOD08_M3 |
| Cloud Fraction | CF | CRU TS4.02 and MOD08_M3 |
| Precipitation | PRE | CRU TS4.02, GPCP, and TRMM |
| 10 m Wind Speed | WSP10 | NCDC-ISD |
| Downwelling Shortwave Radiation (Surface) | SWDOWN | CERES-EBAF |
| Downwelling Longwave Radiation (Surface) | GLW | CERES-EBAF |
| Upwelling Shortwave Radiation (TOA[*]) | SWUPT | CERES-EBAF |
| Upwelling Shortwave Radiation (Surface) | USRS | CERES-EBAF |
| Upwelling Longwave Radiation (TOA[*]) | OLR | CERES-EBAF |
| Shortwave Cloud Forcing | SWCF | CERES-EBAF |
| Longwave Cloud Forcing | LWCF | CERES-EBAF |

[*]: Top of the Atmosphere

---

## Author Comment (AC2) · 25 Mar 2021

**Limitations of WRF land surface models for simulating land use and land cover change in Sub-Saharan Africa and development of an improved model (CLM-AF v. 1.0)**

Timothy Glotfelty, Diana Ramírez-Mejía, Jared Bowden, Adrian Ghilardi, and J. Jason West

**Replies to Referee 2**

In this study, Glotfelty et al. assess the ability of several land surface models to simulate the surface characteristics of sub-Saharan Africa, within the WRF regional numerical weather prediction framework. They find that the default models do a rather poor job in the region, in terms of, for instance, albedo and leaf area index. Consequently, they develop a new CLM land surface model variant, wherein they have improved the representation of such surface properties. Following, they perform a land use/land cover change experiment to highlight the applicability of the new variant, since they show how the meteorological proficiency of the default models hides problems in surface processes, which can lead to flawed conclusions in such experiments. I agree with RC1 that the topic is within the scope of GMD and that this work is relevant not only for the WRF modelling community, but also for land use/land cover change studies focusing on Africa and the Southern Hemisphere. I also find the manuscript to be well outlined, with a clear motivation, sufficient methods, and an in-depth analysis. However, I think it could be improved by addressing the following specific comments. Additionally, I list some technical details that I think should be corrected.

Response: We thank the reviewer for their comments on this work. Our point-by-point responses to the reviewer's comments can be found below.

**Specific comments:**

1) After reading the major comment 1) in RC1, it is clear to me that authors need to improve a few things in Section 4 to better explain their experimental setup. I agree in Section 4.1 they should address RC1 comments, and justify why a single-year validation is enough for the scope of the study, and why it is desirable it is not during an El Nino-Southern Oscillation phase. Maybe a reference to a similar study could help, or stating explicitly the proof-of-concept nature of the validation. Perhaps avoid use the word "validation", in case other studies have validated these LSMs in Africa

Response: We thank the reviewer for their comment. We understand both this reviewer's and the other reviewer's concerns. As mentioned in our response to reviewer 1, the model validation experiment is only meant to serve as an "out-of-the-box" meteorological comparison/evaluation of the WRF LSMs to illustrate the meteorological impacts of the default LSM deficiencies. Our overall goal with this experiment is to assess the impact of each LSMs land surface parameters on the model's meterorological performance. Since these parameters are unchanging in time, the performance of each LSM relative to the others will be similar in time regardless of the meteorological state. Therefore, it is unlikely that running

the model for additional years will change the results in any way significant enough to justify the computational expense. For simplicity we chose 2013 because of its neutral ENSO status, since this is the most common ENSO phase (https://origin.cpc.ncep.noaa.gov/products/analysis_monitoring/ensostuff/ONI_v5.php) and thus would be the most representative single year to simulate.

To address this comment we have changed the name of the model validation experiment to the "Meteorological evaluation experiment" as this is a more accurate description of the goal of this experiment. Additionally, we have added additional descriptions to Section 4.1 to state the proof-of-concept nature of the validation more clearly:

Lines 266-257 revised manuscript:

"The **meteorological evaluation** experiment consists of five simulations conducted for the year 2013, each using one of the five LSM configurations discussed above.  The year 2013 is selected because it is a neutral year for the El Niño Southern Oscillation **(ENSO) and thus should be representative of the mean state of Sub-Saharan Africa's ENSO climate variability**. While a single year comparison does not yield climate relevant statistics, it is sufficient to demonstrate differences in the meteorology between the five LSM configurations and the mechanisms responsible for these differences. **This is because the prescribed surface parameters from the LSM do not vary between years and thus the impact from these parameters on the simulated meteorology will be similar (or at least the impact from each LSM will remain similiar relative to the others) regardless of the model's overall meteorological state.** The **meteorological evaluation** simulations are conducted with default greenhouse gas concentrations and MODIS 21 class land use data. These default settings are chosen to illustrate the performance that can be expected from the publicly available WRF model."

2)  Then I think Sections 4.2 and 4.3 are failing to explain the time period simulated. The comment in RC1 confuses me, because what I understand is that simulations are for boundary conditions of period 2010-2015, all of them, only that in LU01 land use maps are from MODIS 2001, and in LUD land use maps are from Dinamica EGO results. Is this right? If so, then comparison between LU01 and LUD is done subtracting 6-year (2010-2015) averages. Is this right? Authors should explain why this time span was chosen. I think the different time periods in Section 4.3, to explain the Dinamica EGO methods, may confuse the reader. Why mention year 2050? Why add in line 317: "and to 2050 for simulation purposes"?..

Response: We thank the reviewer for their comment. There does appear to be some confusion concerning the land use impact experiments in RC1. What the reviewer indicated is indeed correct. Both simulations (i.e., LU01 and LUD) are simulating the same meteorological period using boundary conditions of 2010 to 2015, LU01 uses MODIS 2001 land use and LUD uses Dinamica EGO land use for each simulated year, and the differences between LU01 and LUD are found by subtracting the 6-year averages.  We have updated the text in Section 4.2 for clarity

and have added a line explaining the reasons why we chose the 2010-2015 period (See underlined below).

Lines 277-291 revised manuscript:

 "The LULCC experiment simulates **recent climate responses from LULCC since the year 2001 by comparing simulations with static LULC from 2001 with dynamic LULC representing 2010-2015.  In both cases, meteorology is simulated for the six-year period of 2010-2015.** **These** two simulations differing in LULCC **are conducted for each LSM configuration, using the Noah, Noah-MP, CLM-D, and CLM-AF LSMs**. The first simulation for each LSM uses static LULC from **MODIS representing** the year 2001 for each simulated year (i.e., 2010-2015), hereafter referred to as LU01. The second uses dynamic LULC **from the MODIS 21 class land use dataset that is processed by the Dinamica EGO land use modeling framework (Soares-Filho et al.,  2002 – described in more detail below)** for each simulated year **in the 2010-2015 period**, hereafter referred to as LUD. The **six-year average** differences between the LU01 and LUD simulations delineate the climate response to LULCC. **The time period 2010-2015 is selected because it is far enough away from the year 2001 to show significant impacts from LULCC and because it contains the full ENSO climate variability cycle.**  Noah-Sat is excluded because LAI and albedo parameters derived from satellite data could be impacted by climatological variability**, and therefore do not only represent** LULCC. The LULCC simulations also utilize global average greenhouse gas concentrations for each simulation year **(2010-2015)** from the National Oceanic and Atmospheric Administration's (NOAA) Earth System Research Laboratory (ESRL) Global Monitoring Division. In the LULCC experiment, each year is a discreet simulation with a 3-month spin-up in which the model LULC is updated at the start of each year. This is necessary because the WRF modelling framework treats LULC as a static field."

The rationale for mentioning the 2050 time horizon in Section 4.3 was to inform the reader that the LULC data used in this study is just a small subset of a much larger dataset. However, since this is causing confusion the mention of this has been removed per the reviewer's suggestion.

3) Minor comment 2) of RC1 : does it perhaps have to do with the fact that SWDOWN is compared with CERES and CF with MODIS?.

Response: While comparisons to two different datasets may play a role, the true reason is that SWDOWN is tied to cloud optical thickness rather than cloud fraction as we have added to the manuscript below:

Lines 465-467 revised manuscript:

"**The underestimated cloud radiative forcing seems to indicate the model is not generating clouds of sufficient optical thickness, since cloud fractions are overestimated compared to satellite estimates (Fig. 10, Fig. S15).**"

4) I think also a table with the data used from ERA-Interim in the Supplementary Material could be helpful.

Response: We thank the reviewer for their comment. Per the reviewer's suggestion, we have added Table S1 to the Supplementary Material.

Table S1: Variables Used from ERA-Interim Reanalysis

| Variable | Units |
|---|---|
| Atmosphere Temperature | K |
| Geopotential Height | $m^2\ s^{-2}$ |
| East -West Wind Component (U) | $m\ s^{-1}$ |
| North -South Wind Component (V) | $m\ s^{-1}$ |
| Relative Humidity | % |
| Surface Pressure | Pascal |
| Sea-Level Pressure | Pascal |
| Surface Skin Temperature | K |
| Sea Surface Temperature | K |
| Soil Temperature | K |
| Soil Moisture | $m^3\ m^{-3}$ |
| Sea Ice Fraction | Fraction |
| Snow Density | $kg\ m^{-3}$ |
| Snow Height | m |

5) I agree with RC1 that the role of Noah-Sat in the study is not explicitly stated. I understand it is used as a proxy for observations in Section 5, and that in Section 6 it helps assess the parameterizations in Noah and Noah-MP, since the surface properties should be ok. But I wonder if Section 2.1.1 could be improved to better explain these reasons why Noah-Sat is included.

Response: We understand both this reviewer and reviewer 1's concerns. We have added the following discussion to Section 2.1.1 to indicate the reasons why Noah-Sat is used per the reviewer's suggestion.

Lines 130-132 revised manuscript:

"**However, Noah-Sat is useful for meteorological evaluations, because it has the most accurate surface parameters in the current WRF modelling system. Therefore, Noah-Sat can be used to understand deficiencies in the surface parameter methodology of the Noah LSM and other WRF LSMs.**"

6) I agree with RC1 that it is easiest to assess performance with maps of differences (e.g., Figs. 2-5, and Figs. 7-9).

Response: We thank the reviewer for their comment. We understand both this reviewer and reviewer 1's concerns. We evaluated model performance in the meteorological evaluation experiment through both soccer goal plots and spatial plots. The soccer goal plots provide insights into the model performance at both a domain-wide and regional level that convey to the reader how far off from the observations the different parameters are. The spatial plots display how well the different model configurations capture the spatial patterns and magnitudes of our three key variables of interest (i.e., USRS, T2, and precipitation). Because we think these elements of the evaluation are the most relevant we have elected to keep these as the model evaluation figures in the main text. However, to assist readers and the reviewer we have also included difference plots of our three key variables in the supplementary material with citations in the main text (See below).

Lines 452-253 revised manuscript:

"**Additionally, annual average difference plots with CERES-EBAF for each LSM are shown in Fig. S6.**"

Lines 469-470 revised manuscript:

"**Annual average differences between CRU and the LSMs are also shown in Fig. S11**."

Lines 472-473 revised manuscript:

"The only clear impact of surface albedo inaccuracy on annual average T2 is the relatively stronger cold bias in the Noah LSM (Fig. 9, **Fig. S11**)."

Lines 507-508 revised manuscript:

"**Additionally, seasonal spatial plots of PRE compared with TRMM and annual average differences between TRMM and the LSMs and shown in Fig. S16 and Fig. S17, respectively.**"

Lines 510-512 revised manuscript:

"The greatest underpredictions occur in arid regions (ND, ED, SD, NESD, and WSD) and portions of East Africa (EM, CM, and LVW), while regions in South Africa (SSD and SM) and EW typically experience the strongest overprediction across the LSMs (Fig. 10, Fig. S12, **Fig. S17**)."

[Figure]

Fig S6: 2013 annual average differences in upwelling shortwave radiation at the Earth's surface (W m⁻²) between the WRF simulations and CERES-EBAF estimates (WRF – CERES-EBAF)

[Figure]

Fig S11: 2013 annual average differences in 2-m temperature (°C) between the WRF simulations and CRU estimates (WRF – CRU)

[Figure]

Fig S17: 2013 annual average differences in precipitation (mm day$^{-1}$) between the WRF simulations and TRMM estimate (WRF – TRMM)

7) Is this the first assessment of all these LSMs in Africa using WRF? If so, state it somewhere. If not, include references.

Response: To our knowledge, this is the first time these LSMs have been examined with this level of detail in Africa. We have added the following line to the introduction to indicate this.

Lines 77-78 revised manuscript:

"**To the authors' knowledge, this is the first time the surface parameters of these LSMs have been robustly assessed is Sub-Saharan Africa**"

8) In the end, it was not clear why the regionalization of Africa occurred. In Section 4.3 I learn that it is for the Dinamica EGO predictions, however in Section 3.2 I was told it was for the LAI monthly profiles. Is it both? If so, connect the two ideas.

Response: We thank the reviewer for their comment and we understand the reviewer's concerns. The regionalization of Africa is a necessary part of the Dinamica EGO modeling system but we adapted it to also make better LAI profiles for CLM-AF. We have updated the text in both Sections 3.2 and 4.3 to connect the two ideas per the reviewer's suggestion.

Section 3.2

Lines 204-212 revised manuscript:

"Since the Sub-Saharan Africa domain covers a wide range of tropical and sub-tropical latitudes, a single domain-wide LAI and SAI monthly profile for each PFT is not appropriate. Here, geographically varying monthly LAI profiles are generated **using 17 distinct regions** based on bioclimate characteristics used in LULCC modelling of Sub-Saharan Africa (Fig.1, Table 3). These bioclimate regions **are constructed for land use modeling purposes as discussed in Section 4.3,** because landscape dynamics are known to be different between broad climatic zones, needing separate modelling parametrizations (Soares-Filho et al 2006). **These same bioclimate regions are ideal for parameterizing LAI and SAI profiles because they divide the region based on climate characteristics that impact vegetation.** However, the central wet (CW), central moist (CM), and northeast semi-dry (NESD) bioclimate regions **used in the land use modeling** span a large latitudinal range and are subdivided based on latitude to generate more meaningful LAI seasonal profiles (Supplementary Material ST2)."

Section 4.3

Lines 333-341 revised manuscript:

"**For Dinamica EGO,** Africa is regionalized into 18 regions based on climatic zones, demographic factors, and anthropogenic activity (Fig. S1) consisting of three overlapping layers: 1) United Nations geographic regions for Africa: Northern, Eastern, Southern, Western and Central (UNSD, 1999); 2) a bioclimate layer from the modified version of the Global Environmental Stratification (GEnS) dataset (Metzger et  al. 2013); and 3) residential sector emissions hotspots using DICE-Africa emissions (Marais and Wiedinmyer, 2016). The resulting 67 categories are generalized into the final 18 based on neighborhood. The process of generalization is done by comparing major change trends among regions, trying to avoid as far as possible the presence of separate regions with similar LULC dynamics. Of these 18 regions, **17 are used in the WRF modelling and for the generation of LAI and SAI profiles as described in Section 3.2** because the North Semi-dry region is outside the Sub-Saharan Africa domain."

9)   The study prefers to use Dinamica EGO output instead of MODIS directly to reduce"noisiness". Is this a usual practice? Is there some previous reference?

Response: We thank the reviewer for their comment. To the authors' knowledge conducting a land use and land cover experiment with annually varying land use is a novel task that we have not seen before in the literature. Therefore, the use of a land use model product for this task rather than a remote sensing product is also a novel practice. The rationale behind this choice is twofold.

First, the LULCC experiment is a "proof-of-concept" experiment in which our goal is to determine if the climate change signals from land use and land cover changes make logical sense with the different LSMs. This can be accomplished with any land use and land cover

dataset that shows a change whether it is realistic or idealized. In our case, these changes are realistic because they are based on observed changes.

Second, we are using the 5th version of the MODIS land use and land cover data, which was the newest product available when this experiment was conducted. This product was only available until the year 2012, so using a model product to represent the years 2013-2015 was only possible using a simulated product. In addition, we found that these MODIS data show frequent transitions between land uses in some grid cells, including transitions back and forth between two land cover classifications.  Our filtering methodology helps to remove these, but using Dinamica EGO enables us to keep only the strongest transitions that have statistical relationships to other land surface properties and socio-economic variables. In this way, Dinamica EGO acts as a secondary filter to only keep the most robust land use transitions.  A 6th version of MODIS that covers the whole period of interest has been released recently, after this project was well underway. However, because the accuracy of the land use product is not the focus of the experiment we do not think redoing the land use and land cover experiments with the updated MODIS land use product is necessary, since it will not affect our findings on how the different LSMs react to land use and land cover changes. We have updated the text in Section 4.3 to make this point clearer.

Lines 301-307 revised manuscript:

 "The LULC dataset for the LULCC experiment are created by means of prospective landscape modelling techniques and while simulations contain some level of model error, this approach is used to reduce the impact of potential LULC misclassification errors and uncertainties in the MODIS product that could propagate into the WRF model leading to "noisy" and inconclusive climate impacts. **To the authors' knowledge, this is a novel practice as many LULC studies in Africa do not simulate year to year changes from the LULC datasets (e.g., Otieno and Anyah, 2012) or use idealized LULC datasets (e.g., Abiodun et al., 2008; Wang et al., 2016). The use of a simulated LULC product is sufficient to support the LULCC experiment, which aims to determine if the climate signals from realistic LULCC simulated by the different LSMs make logical sense.**"

10) How different is the "default" land use in the validation and the MODIS 2001? Is "validation 2013" very different from "year 2013 of LU01 "??

Response: We thank the reviewer for their comment. There are a few non-trivial differences between the WRF default LULC dataset and the MODIS 2001 dataset used in the LU01 experiment. It is not possible for us to track down exactly why they are different because the WRF developers did not keep very detailed records of which MODIS version the default LULC product is. They have indicated that they received these data (i.e., modis_landuse_20class_30s_with_lakes) from the National Centers from Environmental Prediction (NCEP) and that they are likely representative of the year 2001.  See the following link: https://forum.mmm.ucar.edu/phpBB3/viewtopic.php?t=156. We also contacted the WRF developer responsible for static data at the National Center for Atmospheric Research (NCAR), Michael Duda, and he provided the same response at the above link.  We have

added a comparison plot of these two different land use datasets in the Supplementary Material and a brief discussion of the differences in Section 4.2.

[Figure]

Fig S1. Comparison of the default WRF LULC dataset used in the meteorological evaluation simulations and the 2001 MODIS LULC dataset used in the LU01 simulations

Lines 292-299 revised manuscript:

"**There are several non-trivial differences between the WRF default LULC used in the evaluation experiments and the MODIS data used in the LU01 simulation (Fig. S1), even though the WRF default LULC is intended to represent 2001. Overall, the default WRF LULC data has more area classified as grassland, savanna, and forest, with less areas classified as cropland, woody savanna, and barren land compared to the LU01 dataset. Spatially, the areas classified as cropland in LU01 are primarily classified as the nearest natural LULC type in the default dataset. In Central Africa, some areas classified as forests and savannas in the default LULC dataset have been assigned as woody savanna in LU01. In southern Africa, some areas assigned as grasslands in the default LULC dataset are classified as open shrubland and in arid regions some areas classified as open shrubland in the default dataset are assigned as barren land in LU01.**"

11) Is Fig. 1O similar to other results of LULCC studies in Africa?

Response: We thank the reviewer for their comment. It is difficult for us to compare our LULCC with other studies in Africa because they either use idealized LULCC or do not simulate year-to-year change in LULC. However, increased agricultural expansion and deforestation is consistent with at least one of these studies. Thus, we have added the following line to our discussion in Section 7. (Note that Figure 10 is now Fig. 12.)

Lines 545-548 revised manuscript:

"**While it is difficult to compare the LULCC predicted by Dinamica EGO to other African LULCC studies because these studies either use idealized LULCC (e.g., Abiodun et al., 2008; Wang et al., 2016) or do not simulate year-to-year changes, the increased agricultural expansion and deforestation/degradation are consistent with the LULCC seen in Otieno and Anyah (2012) for the period of 1986–2000.**"

12) I do not know how to interpret Table 4. Please explain what is "First Region" and "Second Region".

Response: We thank the reviewer for their comment. The first and second regions are the nearby regions we gather LAI profile information from if a given bioclimate region does not have enough grid cells with that plant functional type to calculate a meaningful LAI profile. We have expanded the discussion to make this point clear.

Lines 225-229 revised manuscript:

"**The first nearby alternative bioclimate region used to generate LAI profiles for the missing PFTs is listed as "First Region" in Table 4. If the "First Region" does not have all the missing PFT LAI profiles then these profiles are obtained from a second nearby bioclimate region ("Second Region").**"

**Technical comments:**

1) As pointed out in RC1, some acronyms are not explained. I point out RL. Also in Table 9 it says WS 10, but in text it is used as WSP10.

Response: We thank the reviewer for their comment. We have fixed several acronym inconsistencies throughout the paper including WSP10 and RL, as well as those used in Table 9.

Table 9: Evaluated Variables and Evaluation Datasets

| Variable | Acronym | Evaluation Dataset |
|---|---|---|
| 2-m Temperature | T2 | CRU TS4.02 and NCDC-ISD |
| Daily Maximum Temperature | T2MAX | CRU TS4.02 |
| Daily Minimum Temperature | T2MIN | CRU TS4.02 |
| Diurnal Temperature Range | DTR | CRU TS4.02 |
| 2-m Vapor Pressure | E2 | CRU TS4.02 |
| 2-m Dew point Temperature | $T_d2$ | NCDC-ISD |
| Precipitable Water Vapor | PWV | MOD08_M3 |
| Cloud Fraction | CF | CRU TS4.02 and MOD08_M3 |
| Precipitation | PRE | CRU TS4.02, GPCP, and TRMM |
| 10 m Wind Speed | WSP10 | NCDC-ISD |
| Downwelling Shortwave Radiation (Surface) | SWDOWN | CERES-EBAF |
| Downwelling Longwave Radiation (Surface) | GLW | CERES-EBAF |
| Upwelling Shortwave Radiation (TOA[*]) | SWUPT | CERES-EBAF |
| Upwelling Shortwave Radiation (Surface) | USRS | CERES-EBAF |
| Upwelling Longwave Radiation (TOA[*]) | OLR | CERES-EBAF |
| Shortwave Cloud Forcing | SWCF | CERES-EBAF |
| Longwave Cloud Forcing | LWCF | CERES-EBAF |

[*]: Top of the Atmosphere

2) Abstract: "climate signals" is used too often. I wonder if "illogical" is the right word.

Response: We thank the reviewer for their comment. We have reworded the abstract to reduce the number of times "climate signals" is used.

We have debated what word best describes the changes deemed illogical and we believe that "illogical" is the best choice. Using the words "incorrect" or "inaccurate" implies that there is an observed change against which these climate responses are compared. This is not the case. What is occurring is that the surface parameters are incorrect which leads to climate changes that are not logically/physically consistent with the given land use and land cover change. Thus, the use of the word "illogical" is more accurate.

Lines 10-27 revised manuscript:

"Land use and land cover change (LULCC) impacts local and regional climates through various biogeophysical processes. Accurate representation of land surface parameters in land surface models (LSMs) is essential to accurately predict these LULCC-induced climate signals. In this work, we test the applicability of the default Noah, Noah-MP, and CLM LSMs in the Weather Research and Forecasting Model (WRF) over Sub-Saharan Africa. We find that the default WRF LSMs do not accurately represent surface albedo, leaf area index, and surface roughness in this region due to various flawed assumptions, including the treatment of the MODIS woody savanna LULC category as closed shrubland. Consequently, we developed a WRF CLM version with more accurate African land surface parameters (CLM-AF), designed such that it can be used to evaluate the influence of LULCC. We evaluate meteorological performance for the default LSMs and CLM-AF against observational datasets, gridded products, and satellite estimates. Further, we conduct LULCC experiments with each LSM to determine if differences in land surface parameters impact the LULCC-induced **climate responses**. Despite clear deficiencies in surface parameters, all LSMs reasonably capture the spatial pattern and magnitude of near surface temperature and precipitation. However in the LULCC experiments, inaccuracies in the default LSMs result in illogical localized temperature and precipitation **changes**. Differences in thermal **changes** between Noah-MP and CLM-AF indicate that the temperature impacts from LULCC are dependent on the sensitivity of evapotranspiration to LULCC in Sub-Saharan Africa. Errors in land surface parameters indicate that the default WRF LSMs considered are not suitable for LULCC experiments in tropical or Southern Hemisphere regions, and that proficient meteorological model performance can mask these issues. We find CLM-AF to be suitable for use in Sub-Saharan Africa LULCC studies, but more work is needed by the WRF community to improve its applicability to other tropical and Southern Hemisphere climates."

3) Text sections in supplementary material should be labelled, to be referred to from the main text easily (e.g., ST1, ST2).

Response: We thanks the reviewer for their comment. The supplementary material text sections have been names ST1-ST4 with references in the text per the reviewer's suggestion.

4) Check the use of "en dash" for ranges of numbers (e.g., domain ranges).

Response: We thank the reviewer for their comment. We have corrected the usage of "en dash" throughout the manuscript and supplementary material.

5) I would stick to "DJF" for winter season, rather than "JFD".

Response: We thank the reviewer for their comment. "JFD" has been changed to DJF throughout both the manuscript and supplementary material.

6) Authors prefer the use of block titles for referring to panels in figures, instead of using labels (e.g., (a), (b)). It is ok, but I think the font size could be reduced. Also figures should use the same font types (titles are serif, but coordinates are sans-serif).

Response: We thank the reviewer for their comment. We like the size of the block title headers because it improves readability. For all panel figures, they are generated via the use

of two programs. The first is NCL (NCAR command language) that generates the individual plots and the second is Adobe Illustrator to assemble them into a large panel figure. The default font of NCL is Helvetica, which is not available in Adobe Illustrator due to licensing issues. Therefore, we have elected to keep the figures as they are because there is no way for us to obtain matching fonts given our current tools without a non-trival amount work to manually reprocess each figure for little benefit to the manuscript.

7) Tables are missing punctuation signs in captions and footnotes. I also think that tables with a lot of text are more readable if left-aligned (raggedright).

Response: We thank the reviewer for their comment. We have added punctuation to all table captions and footnotes. The vast majority of our tables contain acronyms or numbers rather than a lot of text so we have left them all center-aligned for consistency.

8) Sometimes too many references are used. For instance, the number of applications of Dinamica EGO has 12 references.

Response: We thank the reviewer for their comment. To our knowledge, this is the first time Dinamica EGO is being used for an application in a weather/climate model at this scale. For this reason, we want the readers to be aware that this is a robust modeling tool with many different applications. Since the journal does not have any specific rules or page limit restrictions regarding these references we have elected to keep the references for this purpose.

9) Figure captions in supplementary material say "verses" instead of "versus".

Response: We thank the reviewer for their comment. The figure captions have been corrected the captions to say "versus".

10) I wonder if the inset boxes in all maps with extreme and mean values are really being used. I like the idea, but they are not being used in the text or discussion.

Response: We thank the reviewer for their comment. Even though these inset boxes are not being used to a great extent in the text discussions, to make the discussions more succinct, we do think they provide the reader with additional insights and information should the reader be interested. For example, in Fig. 2 the maximum albedo value for Noah-Sat is 45%, which helps the reader see how poorly the default LSMs perform (maximum of 23–28%). Additionally, removing these inset boxes would be a very time consuming non-trivial task that adds little to no benefit to the manuscript. For these reasons, we have elected to keep the figures as they are.

11) Is Table S3 really necessary? Maybe only referencing Friedl, et al. 2002 is enough. By the way, this reference is nowhere in the references list.

Response: We thank the reviewer for their comment and catching this missing reference. We do believe Table S3 is necessary because it is an important piece of the discussion in Section

3.1 and it would be cumbersome for the reader to have to track down this paper to understand the differences in the land cover classes for this discussion. We have added the reference to the reference list.

"Friedl, M., McIver, D. K., Hodges, J.C.F., Zhang, X.Y., Muchoney, D., Strahler, A.H., Woodcock, C.E., Gopal, S., Schneider, A., Cooper, A., Baccini, A., Gao, F., and Schaaf, C.,: Global land cover mapping from MODIS: algorithms and early results, Remote Sensing of Environment, 83, 287-302, https://doi.org/10.1016/S0034-4257(02)00078-0, 2002."

**Line comments:**

We thank the reviewer for their line comments. Our response to each comment can be found below.

1) through A land surface model

Response: The correction has been made:

"Impacts of LULCC are simulated in climate and numerical weather prediction models through a land surface model (LSM)."

2) different SIMULATED climate responses

Response: The correction has been made:

"Differences in LSM parameterizations can lead to significantly different simulated climate responses to LULCC in both magnitude and sign (e.g., Olsen et al., 2004; Boisier et al., 2012; Burakowski et al., 2016), even when little difference exists in the mean simulated climate (Crossly et al., 2000)"

3) and IN the PRESCRIPTION OF

Response: The correction has been made:

"Errors and uncertainties in LSMs occur in response to errors in LULC classification maps and in the prescription of land use properties, such as vegetation distributions and surface albedo (e.g., Lu and Shuttleworth, 2002; Olsen et al., 2004; Ge et al., 2007; Boisier et al., 2012; Boisier et al., 2013; Boysen et al., 2014; Meng et al., 2014; Hartley et al., 2017; Bright et al., 2018)."

4) with WIDESPREAD surface heterogeneity

Response: The correction has been made:

"Having accurate representations of these parameters is especially important in regions with widespread surface heterogeneity, such as East Africa (Ge et al., 2008)."

5) Air Force AND Hydrology Lab (Noah)

Response:  The correction has been made:

"Results from these WRF simulations are somewhat contradictory as some studies found the National Centers for Environmental Prediction, Oregon State University, Air Force and Hydrology Lab (Noah) LSM (Chen and Dudhia, 2001; Ek et al., 2003) to have superior performance compared to observations and reanalyses (Pohl et al., 2011; Igri et al., 2018), while others found no unambiguous difference in model performance between different LSMs (Noble et al., 2014; 2017)."

6)  reanalyses (maybe plural?)

Response:  The correction has been made:

"Results from these WRF simulations are somewhat contradictory as some studies found the National Centers for Environmental Prediction, Oregon State University, Air Force and Hydrology Lab (Noah) LSM (Chen and Dudhia, 2001; Ek et al., 2003) to have superior performance compared to observations and reanalyses (Pohl et al., 2011; Igri et al., 2018), while others found no unambiguous difference in model performance between different LSMs (Noble et al., 2014; 2017)."

7)  parametrization or parameterization?

Response:  It should be parameterization. The correction has been made:

"Physics parameterizations common to all simulations include: the New Tiedtke cumulus parameterization scheme (Zhang et al., 2011), the aerosol-aware Thompson microphysics scheme (Thompson and Eidhammer, 2014), the RRTMG long and shortwave radiation schemes (Clough et al., 2005; Iacono et al., 2008), and the MYNN surface/ planetary boundary layer physics (Nakanishi and Niino, 2004; 2006)"

8)  each year, EVERY

Response:  The correction has been made:

"Because LULC inputs change each year, every model year is simulated individually, preceded by a three-month spin-up period that is discarded to allow the model to reach equilibrium and minimize the impact of initial conditions on the simulations."

9)  model year is SIMULATED

Response:  The correction has been made:

"Because LULC inputs change each year, every model year is simulated individually, preceded by a three-month spin-up period that is discarded to allow the model to reach equilibrium and minimize the impact of initial conditions on the simulations."

10)  why "In"? Maybe only: "Noah and Noah-Sat are"

Response:  It should just be "Noah and Noah-Sat are". The correction has been made:

"Noah and Noah-Sat are the same LSM with different configurations for how surface albedo and LAI are prescribed."

11) also? why also?

Response: This sentence has been restructured in accordance with specific comment 5 above:

"However, Noah-Sat is useful for meteorological evaluations, because it has the most accurate surface parameters in the current WRF modelling system. Therefore, Noah-Sat can be used as pseudo-observations understand deficiencies in the surface parameter methodologies of the other WRF LSMs"

12) "(Sellers 1985)", check citation format

Response: A comma was added for consistency:

[revised manuscript text omitted]

[*]: Top of the Atmosphere